# AGE: Adaptive-masking for Graph Embedding in Graph Retrieval-Augmented Generation

## Abstract

GraphRAG is an extension of retrieval-augmented generation (RAG) that supports large language models (LLMs) by referring to graph-structured data as external knowledge. While this technique ideally captures intricate relationships, it often struggles with graph representations for LLMs, particularly for frozen LLMs, due to the misalignment between graph-based and text-based latent features. We tackle this issue by introducing the *Adaptive-masking for Graph Embedding (AGE)*. AGE employs a Transformer in a mask-based self-supervised learning (SSL) approach. We designed the architecture similar to text embedding encoders, addressing the latent feature misalignment. In contrast to natural language texts, graphs are concise representations, and there exist *key nodes* that hold dominant contextual information, which are challenging to predict from their surroundings. Masking such key nodes leads to inefficiency in the SSL process. Therefore, AGE focuses on predicting nodes apart from key nodes, utilizing a learnable node sampler. Our experimental results indicate that AGE significantly improves approaches using non-parametric search component in GraphQA tasks, achieving superior accuracy across four benchmark datasets with distinct characteristics.

## 1 Introduction

Large Language Models (LLMs) such as GPT-4 OpenAI (2024), Gemini Team et al. (2023), and LLaMA Grattafiori et al. (2024) have significantly advanced natural language understanding and generation capabilities. Retriever-Augmented Generation (RAG) Fan et al. (2024); Gao et al. (2024); Wu et al. (2024a) integrates query-relevant information into the generation process, enabling LLMs to access and utilize domain-specific knowledge beyond their pretraining corpus. However, although RAG enhances LLMs with external data, it may struggle to capture essential structured relationships, reducing search precision and reasoning effectiveness Zeng et al. (2024); Yao et al. (2024). Graph Retriever Augment Generation (GraphRAG) Edge et al. (2024); Mavromatis & Karypis (2024) is a technology that uses graphs to overcome the limitations of RAG. Graph data, represented by nodes (entities) and edges (relationships), clearly presents complex relationships. This provides several benefits, such as facilitating data integration Hu et al. (2024), improving search accuracy Han et al. (2025a); Cao et al. (2025), enhancing inference capabilities Guo et al. (2025); Han et al. (2025b), and reducing hallucinations He et al. (2024). By capturing sub-graphs, the broader context and interconnections within the graph structure can be captured, enabling comprehensive information to be accessed for LLMs enhance the performance in domain-specific tasks.

This study investigates GraphRAG methods that operate within practical computational costs. Fine-tuning LLMs can enhance GraphRAG performance, yet it is resource-intensive. Instead, previous methods often focused on the retrieval module as it is a key factor for GraphRAG performance. Trainable retrievers, such as LLM-based retrievers Sun et al. (2024); Luo et al. (2024) realize a higher retrieval accuracy. However, this strategy still requires significant computational overhead. Non-parametric retrievers He et al. (2024); Yasunaga et al. (2021) are efficient and low-cost but may contain redundant or missing critical nodes, leading to the lack of explicit structural constraints. To maintain practicality, we base our method on non-parametric retrievers with frozen LLM, and improve performance of structural representation by updating the graph embedding module. Embeddings play a crucial role in bridging the gap between retrieved graph data and the LLM input space. Multiple methods He et al. (2024); Perozzi et al. (2024) use graph embedding together with

Figure 1: Overview of GraphRAG with the proposed *Adaptive-masking for Graph Embedding* (AGE) embedding. 1) Retrieval: Find graph elements relevant to the query using a non-parametric process. 2) Subgraph Construction: Extend retrieved graph elements with their adjacencies He et al. (2024). 3) Embedding: Use tokenizer and text embedder for textualized graph and query. Apply AGE for structured relationships of the graph. 4) Inference: Input embeddings into LLM to generate an answer.

textualized graph representation (Fig. 1), indicating that the graph encoder should embed relationships between elements, rather than their individual contents. What is the optimal strategy to achieve such encoding for a frozen LLM? We considered two factors: similarity of the embedding space to the LLM's text encoder and its relationship embedding capability. Since the LLM's text encoder uses mask-based SSL Liu et al. (2019); Reimers & Gurevych (2019); OpenAI (2022), which learns to embed relationships between elements into the embedding space by optimize the reconstruction of masked elements. These factors aim to embed relationships between nodes within the retrieved subgraph into the embedding space by adapting the mask-based SSL with minimal modification.

To realize this intention, we propose a novel embedding strategy, the *Adaptive-masking for Graph Embedding (AGE)*. The architecture of AGE is designed to imitate the general self-supervised text embedding process, while incorporating the Joint-Embedding Predictive Architecture (JEPA) LeCun & Courant (2022), which improves representations in the embedding space by eliminating unnecessary detail reconstruction. Although generative SSL shows promise, the quality of reconstruction relies on the discriminability of input nodes Wei et al. (2022); Chien et al. (2022). Random masking fails on non-discriminative nodes, leading to poor representations Bizeul (2024); Seong & Han (2025). To avoid this, the only major modification is adding a node sampler trained via reinforcement learning (RL) to selectively mask nodes, replacing traditional random node masking with an adaptive approach. The motivation for this approach stems from the fact that, graphs are concise logical structures with minimal redundancy. Hence, some nodes are crucial for maintaining graph integrity; we refer to them as *key nodes*. Our RL-based strategy aims to guide SSL to distinguish the representations of key and auxiliary nodes, encouraging LLMs to identify redundant information within the retrieved graph. The contributions as follows: ① We propose AGE, a novel method that represents retrieved subgraphs via key-node and auxiliary-node embedded by RL-guided mask-based SSL. ② Our study reveals adaptive masking approach's notable effectiveness over random masking within GraphRAG. ③ AGE uses a non-parametric retriever and open LLMs, while also achieving SOTA on three other benchmarks.

## 2 RELATED WORK

### 2.1 GRAPH REPRESENTATION FOR LLMS

In the context of representing graphs as input to LLMs, it is necessary to first convert the retrieved graph data into specific formats. We summarize two distinct formats: textualization and graph embeddings. Textualization Fatemi et al. (2024); Jin et al. (2024); Li et al. (2024) is a text-based formalization method designed to characterize and represent graph data. Node sequences are a popular form of textualizationChen et al. (2024b); Luo et al. (2024); Sun et al. (2024). Some methods Luo et al. (2024); Sun et al. (2024); Chen et al. (2024a) propose LLM-based retrievers to extract reasoning paths. A node sequence ordered along the path aids LLM's reasoning. However, many studies

report negative conclusions in interpreting text-encoded graphs with concurrent LLMs Huang et al. (2023); Guo et al. (2023); Wang et al. (2023), suggesting a need for solutions beyond textualization.

The other format, graph embeddings, have recently been adopted in GraphToken Perozzi et al. (2024). Following this, G-Retriever He et al. (2024) proposed a retrieval framework for graph embeddings. This occurs when graph embeddings are added as tunable prompts to the LLM in addition to their textualized representations. In this work, we improve the quality of LLM responses on the G-Retriever framework through enhancing the representation of graph embeddings. Some methods Xu et al. (2025); Hu et al. (2024) build self-alignment and cross-question module among retrieved entities, relations, and subgraph embedding elements. Some methods enhance embeddings through a two-stage training process Ji et al. (2024); Wang et al. (2024). The first stage trains the embedding module on SSL alone; in the second stage, prompt tuning aligns the structured relationships embedded for LLM input by the pretrained module. Since each LLM has its own domain embeddings and input spaces, two-stage training process prioritize maximizing performance. Instead, focusing on practical, we propose a one-stage training process SSL that integrates with prompt tuning.

## 2.2 Self-Supervised Learning

Many existing self-supervised learning architectures focus on learning representations that effectively capture relationships between input data. Joint-Embedding Architecture (JEA) Bardes et al. (2021); Caron et al. (2020); Grill et al. (2020) has shown considerable promise in advancing SSL methodologies. Joint-Embedding SSL for GNNs, such as GraphCL Ying et al. (2021), GCA Zhu et al. (2021) and JOAO You et al. (2021), learn node representations by contrasting positive and negative samples. Subsequent studies identified areas for enhancement in JEA Chin et al. (2024); Jing et al. (2021); Lee et al. (2025), particularly the issue of mapping all inputs to a single constant vector, known as the collapsing problem. Generative Architecture (GA) He et al. (2021); Baevski et al. (2022); Devlin et al. (2018) focuses on reconstructing masked portions of the input at either the pixel or token level. GraphMAE Hou et al. (2022; 2023) learns representations by reconstructing masked samples. These methods encourage the model to learn more robust and diverse representations, potentially reducing the risk of representation collapse Chin et al. (2024); Jing et al. (2021); Assran et al. (2023). Joint-Embedding Predictive Architectur (JEPA) LeCun & Courant (2022) eliminate reconstruction of pixel or token-level details and enhances the semantic level of self-supervised representations Assran et al. (2023); Bardes et al. (2024). In this work, we first demonstrate JEPA's effectiveness in the GraphRAG framework. JEPA originates from cognitive neuroscience, suggesting that humans have an ability of top-down schema reasoning, aiding planning, decision-making, and problem-solving on complex tasks Tang et al. (2007); Mittal et al. (2020); Theves et al. (2021). In GraphRAG, the tokens fed into the LLM's hidden layer should capture the ability. We implement it as JEPA in the graph embedding module. Cognitive science has revealed that a brain region called the temporal lobe plays a role in bottom-up associative learning, selecting key knowledge and linking it to related data Jackson et al. (2018); Edmonds et al. (2019); Cox et al. (2024). These selection processes can also occur with new information. Aiming to reproduce this functionality in GraphRAG, AGE has the novel node sampler module.

## 3 Preliminaries

**GQA with LLM.** For a query $q$ on a textual graph $G$, there is an optimal subgraph $\overline{S^*} \in S(G)$ and query relevant text-modal knowledge $T^*$ that guides the LLM to produce expected answers, where $S(G)$ is the set of all subgraphs of $G$. The challenge of GraphRAG is to efficiently search for the relevant subgraph $S^*$ and represent it to $\overline{S^*}$ for an LLM $p_\Phi$ improve generation. The probability distribution of the output sequence $Y$ is given by:

$$p_\Phi(Y \mid [q, G]) = \prod_{i=1}^{n} p_\Phi(y_i \mid y_{<i}, [q, T^*, \overline{S^*}]), (1)$$

where $y_{<i}$ represents the prefix tokens, and $[q, \overline{S^*}]$ indicates the concatenation of the query, relevant text-modal knowledge and optimal subgraph information, respectively.

**Joint-Embedding Predictive Architecture.** Mask-based SSL methods such as MAE are well suited to handle corrupted input, as they learn to reconstruct missing or corrupted input parts. JEPA improves upon MAE by eliminating the reconstruction of unnecessary input feature details, focusing

instead on learning more abstract representations. JEPA consists of an encoder $E_\theta(\cdot)$, predictor $P_\phi(\cdot)$ and target encoder $T_\theta(\cdot)$. The stop-gradient operation sg is employed to prevent representation collapse in the target encoder $T_\theta(\cdot)$. The predictor generates $y$ from visible input $x$ and masked input $\Delta_x$. The encoder and predictor are trained simultaneously with the objective:

$$\min ||P_\phi\big(\Delta_x, E_\theta(x)\big) - \text{sg}\big(T_\theta(y)\big)||_2, \tag{2}$$

The loss is applied only to the predictions of the masked input $\Delta_x$.

**Reinforcement Learning.** To estimate the key and auxiliary nodes on retrieved graph for mask-based SSL discriminative embedding. We adopt REINFORCE Sutton et al. (2000), a basic policy gradient method in RL. Let $\mathcal{D} = (q, Y^*)$ denote a corpus of training data, where $Y^*$ is the complete reference label for query $q$. REINFORCE optimizes a policy $\pi_\theta$ parameterized by $\theta$, to maximize reconstruction quality $R_a$ for each masking action $a$. The policy gradient is given by:

$$\nabla_\theta \mathcal{J}(\theta) = \mathbb{E}_{q \sim \mathcal{D}, S^* \sim q} \left[ \sum_{v \in S^*} \nabla \log \pi_\theta(a|v) \cdot R_a \right] \tag{3}$$

where where $\mathcal{J}(\theta)$ is the expected return. $\pi_\theta(a|v)$ denotes the probability distribution estimated by policy $\pi_\theta$ for taking action $a$ (masking or not) on node $v$. The SSL framework with masked node reconstruction serves as the RL environment.

## 4 APPROACH

Our framework, illustrated in Figure 1, consists of four main steps: *input*, *graph preprocessing*, *embedding*, and *inference*. We adopt the previous method He et al. (2024) that applies SentenceBert Reimers & Gurevych (2019) to indexed knowledge data at (*input*) step and employ a static k-nearest neighbors Kramer (2013) retrieval approach combined with Prize-Collecting Steiner Tree Bienstock et al. (1993) subgraph construction during *graph preprocessing*. For *inference*, we can use arbitrary LLMs, as usual RAG methods. Therefore, this section focuses on the details of *embedding* step.

### 4.1 TEXT EMBEDDING OF QUERY AND TEXT GRAPH

We transform the retrieved subgraph $S^*$ into a textual format, following He et al. (2024) as in the first two steps. The converted text is then concatenated with the input query $q$. The concatenated texts are embedded into $h_\text{text}$ using a pretrained function for the frozen LLM, TextEmbedding, where $[;]$ denotes concatenation and $L$ is the output token sequence length, as follows:

$$h_\text{text} = \text{TextEmbedding}([\text{textualize}(S^*); q]) \in \mathbb{R}^{L \times d_l}, \tag{4}$$

### 4.2 ADAPTIVE-MASKING FOR GRAPH EMBEDDING

AGE comprises a *node sampler*, *concept encoder-decoder*, *target encoder*, and *graph-structure-based aggregator* modules, as overviewed in Fig. 2. The AGE's input, $h_\text{in}$, is encoded from the retrieved graph $S^*$ with a conventional graph encoder. $h_\text{in}$ is then passed to *node sampler* and *target encoder*. The node sampler categorizes the nodes into *key nodes* and the remaining *auxiliary nodes*. The selected *key nodes* are fed to *concept encoder-decoder*. The output $h_\text{out}$ is trained to predict $h_\text{target}$, the output of *target encoder*, forming a JEPA. The embedding $h_\text{out}$ is then aggregated into a token to be fed to the LLM. The rest of this subsection explains each module in Fig. 2 individually.

**Graph Encoder** prepares input for AGE. The retrieved subgraph $S^* = (V^*, E^*)$ consists of query-relevant nodes $V^*$ and edges $E^*$. $h_\text{in}$ is obtained from $S^*$ by the graph encoder $\text{GNN}_\text{GE}$ as follows:

$$h_\text{in} = \text{GNN}_\text{GE}(S^*; \theta_\text{GE}) \in \mathbb{R}^{N \times d_g}, \tag{5}$$

where $\theta_\text{GE}$ is parameter of $\text{GNN}_\text{GE}$, $d_g$ is dimension of each output node feature, and $N = |V^*|$.

**Node sampler** estimates key nodes using $h_\text{in}$ from the graph encoder for adapting masks on auxiliary nodes. The node sampler processes $h_\text{in}$ through a Multi-Head Attention (MHA) network, a linear layer, and a softmax activation, to obtain nodes probability scores $p_\text{NS}$ as follows:

$$z_\text{NS} = \text{MHA}_\text{NS}(h_\text{in}; \theta_\text{NS}^\text{MHA}) \in \mathbb{R}^{N \times d_g}, \tag{6}$$

$$p_\text{NS} = \text{Softmax}(\text{Linear}(z_\text{NS}; \theta_\text{NS}^\text{Linear})) \in [0, 1]^{N \times 1}, \tag{7}$$

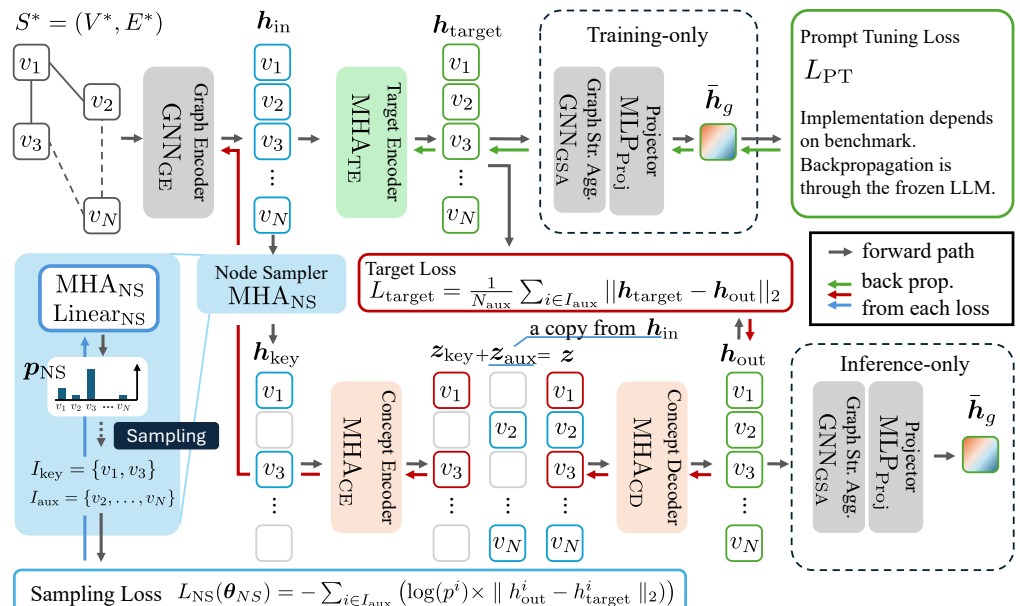

Figure 2: Architecture for Adaptive-masking for Graph Embedding: During training, $h_{\text{target}}$ is connected to the downstream for the target encoder training, while $h_{\text{out}}$ is used during inference. The node sampler explores the optimal distribution for mask-based SSL for graphs. The loss functions train distinct sets of modules without overlap.

where $\boldsymbol{\theta}_{\text{NS}} = \{\boldsymbol{\theta}_{\text{NS}}^{\text{MHA}}, \boldsymbol{\theta}_{\text{NS}}^{\text{Linear}}\}$ is the parameter of this module. We sample $N_{\text{key}}$ nodes based on a categorical distribution defined by $\boldsymbol{p}_{\text{NS}}$, where we decide $N_{\text{key}}$ by the sampling rate $\rho$ as $N_{\text{key}} = \lceil \rho N \rceil$. Hereafter, we denote the sampled key nodes as $I_{\text{key}}$ and auxiliary nodes as $I_{\text{aux}}(= V^* \backslash I_{\text{key}})$. Based on $I_{\text{key}}$, we extract key node features $h_{\text{key}} \in \mathbb{R}^{k \times d_g}$ from $h_{\text{in}}$ and input it to our concept encoder-decoder module.

**Concept Encoder-Decoder** consists of a concept encoder MHA$_{\text{CE}}$ and a concept decoder MHA$_{\text{CD}}$. MHA$_{\text{CE}}$ encode the input $h_{\text{key}}$ into the latent representation $z_{\text{key}}$ as follows:

$$z_{\text{key}} = \text{MHA}_{\text{CE}}(h_{\text{key}} + \text{PE}(h_{\text{key}}); \boldsymbol{\theta}_{\text{CE}}) \in \mathbb{R}^{k \times d_g}, \tag{8}$$

where $\text{PE}(\cdot)$ represents positional encoding as defined by Ma et al. (2021), and $\boldsymbol{\theta}_{\text{CE}}$ is the parameter of MHA$_{\text{CE}}$. $z_{\text{key}}$ is combined with $z_{\text{aux}}$, placeholder vectors for unsampled auxiliary nodes with values copied from $h_{\text{in}}$, as in Zheng et al. (2025). Let $z \in \mathbb{R}^{N \times d_g}$ be the combined node features, which maintains $h_{\text{in}}$'s original node position. MHA$_{\text{CD}}$ decodes $h_{\text{out}}$ with the positional encoding PE as follows:

$$h_{\text{out}} = \text{MHA}_{\text{CD}}(z + \text{PE}(z); \boldsymbol{\theta}_{\text{CD}}) \in \mathbb{R}^{N \times d_g}, \tag{9}$$

where $\boldsymbol{\theta}_{\text{CD}}$ is the parameter of MHA$_{\text{CD}}$. $h_{\text{out}}$ is the output of AGE, which we train to predict $h_{\text{target}}$, an embedding obtained from all nodes through the target encoder.

**Target Encoder** is applied to obtain a prediction target for the previous module in a semantic space (JEPA), as it works more robustly than in the input space (GA) Assran et al. (2023); Chen et al. (2025); Fei et al. (2023); Bardes et al. (2024). The target encoder MHA$_{\text{TE}}$ projects $h_{\text{in}}$ to a target embedding $h_{\text{target}}$ as follows:

$$h_{\text{target}} = \text{MHA}_{\text{TE}}(h_{\text{in}} + \text{PE}(h_{\text{in}}); \boldsymbol{\theta}_{\text{TE}}) \in \mathbb{R}^{N \times d_g}, \tag{10}$$

where $\boldsymbol{\theta}_{\text{TE}}$ is the parameter of MHA$_{\text{TE}}$. We train MHA$_{\text{TE}}$ with downstream tasks, optimizing it to produce embeddings that directly contribute to the task. MHA$_{\text{CE}}$ and MHA$_{\text{CD}}$ are trained in parallel with MHA$_{\text{TE}}$, with the target encoder learning graph representations for LLMs and the concept encoder-decoder learn to exploit key-node representations for mimic the target encoder's auxiliary node representations. Inferring masked auxiliary nodes condenses relational concepts between key and auxiliary nodes into $z_{\text{key}}$ (and thus $h_{\text{out}}$ in the downstream). This JEPA-derived mechanism

would be synergetic with GraphRAG as the previous studies suffers from embedding graph's structured relationships efficiently Huang et al. (2023).

**Graph-structure-based Aggregator** $\text{GNN}_{\text{GSA}}$ projects $\boldsymbol{h}_{\text{out}}$ to a single token $\bar{\boldsymbol{h}}_g$. As the graph encoder module, we followed He et al. (2024) for this module. It aggregates $\boldsymbol{h}_{\text{out}}$ referring $E^*$, the edge connections of the original subgraph $S^*$. Where POOL is mean pooling and $d_l$ is the input dimension of the target layer. The projector $\text{MLP}_{\text{Proj}}$ adjusts the aggregated embeddings to fit the LLM's input dimension.

$$\boldsymbol{h}_g = \text{POOL}(\text{GNN}_{\text{GSA}}(\boldsymbol{h}_{\text{out}}; \boldsymbol{\theta}_{\text{GSA}})) \in \mathbb{R}^{d_g}, \tag{11}$$

$$\bar{\boldsymbol{h}}_g = \text{MLP}_{\text{Proj}}(\boldsymbol{h}_g; \boldsymbol{\theta}_{\text{Proj}}) \in \mathbb{R}^{d_l}, \tag{12}$$

### 4.3 Optimization of Adaptive-masking for Graph Embedding

This subsection describes three loss functions used in our method. In the training phase, we connect $\boldsymbol{h}_{\text{target}}$ in the target encoder stream to the LLM and optimize $\boldsymbol{\theta}_{\text{TE}}$ with the prompt tuning loss $L_{\text{PT}}$. During training the target encoder with $L_{\text{PT}}$, the concept encoder-decoder module is optimized exclusively with $L_{\text{target}}$, in a JEPA approach. Once entire network has been trained, we connect $\boldsymbol{h}_{\text{out}}$ to the downstream LLM rather than $\boldsymbol{h}_{\text{target}}$ at the inference phase.

One challenge of AGE lies in optimizing $\boldsymbol{\theta}_{\text{NS}}$. Optimizing $\boldsymbol{\theta}_{\text{NS}}$ using $L_{\text{target}}$ is difficult due to the non-differentiability of the sampling operation. Therefore, we propose an additional loss function $L_{\text{NS}}$ for optimizing $\boldsymbol{\theta}_{\text{NS}}$.

**Prompt tuning loss** $L_{\text{PT}}$ maximizes accuracy of a downstream task. It was originally introduced in He et al. (2024), and we use the definition as is. We optimize $\boldsymbol{\theta}_{\text{TE}}$, $\boldsymbol{\theta}_{\text{GSA}}$, and $\boldsymbol{\theta}_{\text{Proj}}$ with $L_{\text{PT}}$. The concrete implementation depends on the benchmark tasks; refer to the original papers for details. Note that we train $\boldsymbol{\theta}_{\text{GE}}$ with $L_{\text{target}}$ rather than $L_{\text{PT}}$, as the concept encoder-decoder is used in the inference phase and the upstream network should be optimized to that module.

**Target loss** $L_{\textbf{target}}$ optimizes parameters $\boldsymbol{\theta}_{\text{GE}}$, $\boldsymbol{\theta}_{\text{CE}}$, and $\boldsymbol{\theta}_{\text{CD}}$ to maximize embedding reconstruction by minimizing the distance between $\boldsymbol{h}_{\text{out}}$ and $\boldsymbol{h}_{\text{target}}$ for each auxiliary node indexed by $I_{\text{aux}}$ as follows:

$$L_{\text{target}}(\boldsymbol{\theta}_{\text{GE}}, \boldsymbol{\theta}_{\text{CE}}, \boldsymbol{\theta}_{\text{CD}}) = \frac{1}{N_{\text{aux}}} \sum_{i \in I_{\text{aux}}} \| h_{\text{out}}^i - sg(h_{\text{target}}^i) \|_2, \tag{13}$$

with $N_{\text{aux}}$ defined as $N - N_{\text{key}}$. The objective is to apply knowledge distillation to effectively represent key nodes for reconstructing auxiliary nodes. Furthermore, we apply normalization to enhance stability during the learning process, details are provided in the Appendix.

**Sampling loss** $L_{\textbf{NS}}$ optimizes $\boldsymbol{\theta}_{\text{NS}}$ using RL-inspired supervision. Regarding the operation as an action, the node sampler as a policy network, and $\boldsymbol{h}_{\text{out}}$ as a state, we design $L_{\text{NS}}$ on $\boldsymbol{p}_{\text{NS}} = \{p^1, \ldots, p^N\}$ in Eq. 7 as follows:

$$L_{\text{NS}}(\boldsymbol{\theta}_{\text{NS}}) = -\frac{1}{N_{\text{aux}}} \sum_{i \in I_{\text{aux}}} \left( \log(p^i) \times sg(\| h_{\text{out}}^i - h_{\text{target}}^i \|_2) \right). \tag{14}$$

The loss is back-propagated only to $\boldsymbol{\theta}_{\text{NS}}$. Here, for each node assigned to $I_{\text{aux}}$, larger $\| h_{\text{out}}^i - h_{\text{target}}^i \|_2$ increases $p^i$ more, resulting in pushing such node into $I_{\text{key}}$. This reflects our aim to classify nodes that are difficult to predict from their surrounding as *key nodes*. Additional strategies for key node selections and sampling optimization for RL are discussed in the Appendix.

**Total loss** is given as $L_{\text{PT}} + L_{\text{target}} + L_{\text{NS}}$. Since we designed the optimization process so that each loss optimizes different modules without overlap, our methods does not require weight adjustment between these losses.

### 4.4 Analysis of Learning Objectives

To explain the motivation for architecture design and applying a distributed loss to each module, we analyze the learning objective of the $R\omega$ module, which represents expected $\overline{S^*}$ for LLM $\pi_\theta$ on LoRA finetuning as:

$$\mathcal{L} = \underbrace{-\mathbb{E}_{(S^*)}\left[\log R\omega(\overline{S^*} \mid S^*)\right]}_{\text{Loss of Graph Representation Module}} \underbrace{-\mathbb{E}_{(q, T^*, \overline{S^*})}\left[\log \pi_\theta(i \mid q, T^*, \overline{S^*})\, \pi_\theta(r \mid q, T^*, \overline{S^*}, i)\right]}_{\text{Loss of LLM}} \tag{15}$$

According to Bayes' Theorem, given an input $X$, a target $Y$, and latent rationales $Z$, we can sample these latent rationales $Z$ from the posterior distribution $P(Z|X,Y)$. This posterior represents the probability of latent $Z$ given both the input $X$ and the target $Y$. To compute the marginal likelihood of obtaining answer $Y$ given input $X$, we marginalize over all possible rationales $Z$:

$$P(Y|X) = \sum_{Z \sim P(Z|X,Y)} P(Z,Y|X) \tag{a}$$

$$= \sum_{Z \sim P(Z|X,Y)} P(Z|X) \cdot P(Y|X,Z) \tag{b}$$

The equations above show how to compute the marginal likelihood $P(Y|X)$. Equation (a) makes explicit that $Z$ is sampled from the posterior distribution $P(Z|X,Y)$. Equation (b) applies the chain rule of probability to decompose $P(Z,Y|X)$ into two components: $P(Z|X)$ and $P(Y|X,Z)$. Following this analysis, we apply it to the learning objective for target representation $\overline{S^*}$ given input $S$, latent $\mathcal{Z}$ from a posterior $R\omega(\mathcal{Z}|S,\overline{S})$ that bridges $S$ and $\overline{S}$. The marginal likelihood of $\overline{S}$ given $S$ is:

$$R\omega(\overline{S^*}|S^*) = \sum_{Z \sim R\omega(Z|S^*,\overline{S^*})} R\omega(Z,\overline{S^*}|S^*)$$
$$= \sum_{Z \sim R\omega(Z|S^*,\overline{S^*})} R\omega(Z|S^*) \cdot R\omega(\overline{S^*}|S^*,Z) \tag{16}$$

Above analysis shows that learning objective Graph Representation implicitly learns to identify the latent $Z$ and map it to the expected $\overline{S^*}$ for LLM. The extension of the loss function is:

$$-\mathbb{E}\big[\log_{R\omega}(\overline{S^*} \mid S^*)\big] = \underbrace{-\mathbb{E}\big[\log R\omega(\mathcal{Z} \mid S^*,\overline{S^*})\big]}_{\text{Loss of Latent Identification}} \underbrace{-\mathbb{E}\big[\log R\omega(\mathcal{Z} \mid S^*) \cdot R\omega(\overline{S^*} \mid S^*,\mathcal{Z}))\big]}_{\text{Loss of Representation}} \tag{17}$$

Instead of using a single model for both latent $\mathcal{Z}$ identification and $\overline{S^*}$ representation learning. We explicitly separate the learning into $Sampler_\theta$ for latent identification and $Encoder_\theta$-$Decoder_\theta$ for representation as:

$$-\mathbb{E}\big[\log_{R\omega}(\overline{S^*} \mid S^*)\big] \approx \underbrace{-\mathbb{E}_{(S^*,\overline{S^*})}\big[V_{key} \in \mathcal{Z} \sim \log \mathrm{Sampler}_\theta(\mathcal{Z} \mid S^*,\overline{S^*})\big]}_{\text{Loss of Node Sampling}}$$
$$\underbrace{-\mathbb{E}_{(S^*)}\big[\log Encoder_\theta(\mathcal{Z} \mid V_{key}) \cdot Decoder_\theta(\overline{S^*} \mid \Delta_{V_{masked}}, \mathcal{Z})\big]}_{\text{Loss of Encoder-Decoder}} \tag{18}$$

By separating the learning processes, the target encoder learns the representation directly, while the encoder-decoder learns to reconstruct this representation through **Evidence Lower Bound (ELBO) optimization**. Specifically, the node sampler learns to extrapolate $V_{key} \in \mathcal{Z}$, making static sampling illogical. Therefore, our node sampler with encoder-decoder architecture and explicit loss distribution yields efficient learning signals, faster convergence, and improved graph representations. To support our analysis, we provide empirical comparisons of sampling strategies in Appendix B.3.7 and analyze the stability of the target encoder teacher module for the encoder-decoder in Appendix C.3.13. In the prompt tuning setting, given $r$ as the reasoning trajectory, the learning objective for frozen LLMs with graph representation model $R_\omega$ is:

$$\mathcal{L} = \underbrace{-\mathbb{E}\big[\log R_\omega(\mathcal{Z} \mid S^*,\overline{S^*})\big]}_{\text{Loss of Latent Identification}} \underbrace{-\mathbb{E}\big[\log R_\omega(\mathcal{Z} \mid S^*) \cdot R_\omega(\overline{S^*} \mid S^*,\mathcal{Z}))\big]}_{\text{Loss of Representation}}$$
$$\underbrace{-\mathbb{E}\big[\log \pi_\theta(i \mid q, T^*, \overline{S^*})\big]}_{\text{Loss of Knowledge Recalling}} \underbrace{-\mathbb{E}\big[\log \pi_\theta(r \mid q, T^*, \overline{S^*}, i)\big]}_{\text{Loss of Contextualized Reasoning}} \tag{19}$$
$$\underbrace{\phantom{-\mathbb{E}\big[\log \pi_\theta(i \mid q, T^*, \overline{S^*})\big] -\mathbb{E}\big[\log \pi_\theta(r \mid q, T^*, \overline{S^*}, i)\big]}}_{\text{Frozen}}$$

We observe that $R_\omega$ explicitly learns to identify the latent $\mathcal{Z}$ from $S^*$ for LLM-expected $\overline{S^*}$. During training with frozen LLM parameters, $R_\omega$ implicitly captures latent identification $i$ by satisfying the frozen LLM's expectations: $Retriever(S^* \mid q, T^*)$ $R_\omega(\overline{S^*} \mid S^*, \mathcal{Z}) \approx \pi_\theta(\overline{S^*} \mid q, T^*, i)$, yielding $\mathcal{Z} \subseteq i$. Therefore, $R_\omega$ able to learns a subspace of the frozen LLM's complete latent space through this objective. Throughout this, we argue that leveraging a learned latent space $\mathcal{Z}$, robustly restructured into the LLM-expected representation $\overline{S^*}$, can directly improve knowledge recall and indirectly enhance reasoning.

Table 1: Performance comparison across ExplaGraphs, SceneGraphs, and WebQSP datasets under the five settings. The best and second-best scores are highlighted in **bold** and underline, respectively.

| Setting | Method | LLM | Expla Graphs | Scene Graphs | WebQSP | CWQ |
|---|---|---|---|---|---|---|
| Frozen LLM w/ Graph Embedding | G-Retriever | Llama3.2-1B | 0.5595 | 0.7540 | 60.1 | — |
| | G-Retriever | Llama3.2-3B | 0.7761 | 0.8229 | 71.3 | — |
| | G-Retriever | Llama2-7B | 0.8516 | 0.8131 | 68.1 | — |
| | **AGE G-Retriever** | Llama3.2-1B | 0.8267 | 0.8184 | 62.5 | — |
| | **AGE G-Retriever** | Llama3.2-3B | 0.9260 | 0.8930 | 73.5 | — |
| | **AGE G-Retriever** | Llama3.1-8B | 0.9350 | 0.9276 | 78.3 | — |
| Frozen LLM w/ Graph Embedding + PEFT | G-Retriever | Llama3.2-1B-LoRA | 0.7328 | 0.8689 | 65.3 | — |
| | G-Retriever | Llama3.2-3B-LoRA | 0.8339 | 0.9074 | 71.4 | — |
| | G-Retriever | Llama2-7B-LoRA | 0.8705 | 0.8683 | 70.2 | — |
| | **AGE G-Retriever** | Llama3.2-1B-LoRA | 0.8501 | 0.9056 | 69.1 | — |
| | **AGE G-Retriever** | Llama3.2-3B-LoRA | 0.9134 | **0.9486** | 77.3 | — |
| | **AGE G-Retriever** | Llama3.1-8B-LoRA | **0.9612** | 0.9325 | 80.3 | — |
| | AMAR | Llama2-7B-LoRA | — | — | 84.3 | 82.9 |
| | AMAR | Llama2-13B-LoRA | — | — | 83.3 | 83.1 |
| | **AGE AMAR** | Llama2-7B-LoRA | — | — | 86.5 | **85.2** |
| | **AGE AMAR** | Llama2-13B-LoRA | — | — | 86.2 | 85.1 |
| LLM w/ LLM Retriever | ToG | GPT-4 | — | — | 82.6 | 67.6 |
| | ReKnoS | GPT-4 | — | — | 84.9 | 68.2 |
| | Paths-over-Graph | GPT-4 | — | — | **96.7** | 81.4 |
| | Plan-on-Graph | GPT-4 | — | — | 87.3 | 75.0 |
| | DoG | GPT-4 | — | — | 91.0 | 56.0 |

## 5 EXPERIMENTS

**Datasets and Evaluation Metrics**. Following previous work He et al. (2024); Hu et al. (2024); Ji et al. (2024), we conduct experiments on ExplaGraphs Saha et al. (2021) is a generative commonsense reasoning dataset, SceneGraphs Hudson & Manning (2019) is a visual question answering dataset. And WebQSP tau Yih et al. (2016), ComplexWebQuestions (CWQ) Talmor & Berant (2018) is a large question-answering dataset derived from Web questions, where all queries can be answered using Freebase, a large collaborative knowledge graph database. We use accuracy as the primary metric for ExplaGraphs and SceneGraphs, datasets focusing on reasoning, following He et al. (2024); Hu et al. (2024); Ji et al. (2024). For WebQSP and CWQ, a dataset with extra-large graphs, we use the Hit@1 metric, as in Luo et al. (2024).

**Implementation Details**. We employed the open-source Llama3.2 (1B and 3B) AI (2024), Llama 2 (7b and 13B) Touvron et al. (2023) and Llama3.1 (8B) Grattafiori et al. (2024) as frozen LLM components. Based on the analysis in 5.2, we set the sampling rate $\rho = 0.3$.

### 5.1 MAIN RESULTS

Table 1 illustrates our main results, comparing the methods in three settings: **Frozen LLM with Graph Embedding**: Use a graph embedding technique to tune tokens with given prompt. **Frozen LLM with Graph Embedding + PEFT**: Apply LoRA Hu et al. (2021), a PEFT technique, in combination with the graph embedding technique. **LLM with LLM-Retriever**: Use LLM for retrieval in addition to the one for inference. Any methods train either LLM. These are reference scores with a larger computational cost. Among **Frozen LLM with Graph Embedding** settings (with and without PEFT), **AGE consistently improved performance of G-Retriever and AMAR regardless of the backbone LLM models**. Without PEFT, Llama3.2-1B with AGE showed the most notable gain against G-Retiever: 26.72 percent points increase on ExplaGraphs, while the least gain was observed with Llama3.2-3B on WebQSP, which was 2.02 points. This might be due to the extra-large size of knowledge graphs in WebQSP datasets, which include textual knowledge absent at pretraining, and non-parameter retriever struggling to provide critical information for representation. AGE maintains consistent superiority against G-Retriever and shows more gains on retrieval from smaller graphs. By employing a cross-question approach enriched with retrieved elements, GRAG improved

Table 2: Performance improvements (with Llama3.2 1b, on ExplaGraphs, % pts: percent points).

|  | G-Retriever | GA w Random mask | JEPA w Random mask | JEPA w Node sampler |
|---|---|---|---|---|
| Loss | $L_{PT}$ | $L_{PT} + L_{target}$ | $L_{PT} + L_{target}$ | $L_{PT} + L_{target} + L_{NS}$ |
| Acc | 0.5595 | 0.6532 (↑ 9.37%) | 0.7141 (↑ 15.46%) | 0.8267 (↑ 26.72%) |

performance by 2.8 points, AMAR achieved an improvement of 4.2 points with its baseline. This suggests that improving embedding module is beneficial, it alone may not be enough to boost performance significantly, and relying on a non-parameter retriever could be limiting. Despite that challenge, when integrated with AMAR, AGE continues to achieve further enhancements the performance. Direct comparison with current **LLM with LLM Retriever** methods on WebQSP and the larger CWQ shows that AGE AMAR, which applies non-parametric retriever-based approaches, outperforms the proprietary LLM-based retriever ReKnoS Wang et al. (2025) on both datasets. AGE AMAR underperforms compared with Paths-over-Graph Tan et al. (2025), Plan-on-Graph Wu et al. (2024b) and DoG Ma et al. (2025) on WebQSP, but outperforms them on CWQ, suggesting that AGE AMAR is advantageous on larger datasets, showing substantial potential for future work.

## 5.2 ABLATION STUDY

**Performance Comparison of Self-Supervised Learning Architectures.** Table 2 presents the performance of concept encoder-decoder with some variations and the baseline of G-Retriever, which demonstrates contribution of proposed technique independently. As a SSL variation, we prepared AGE based on generative architecture (GA) against our choice of JEPA. We also compared AGE with random mask against the proposed learnable node sampler. All the results used Llama3.2 1B as its LLM. Using a GA with a random mask, AGE achieves a performance of $0.6532$, which is a $9.37\%$ improvement over the baseline. Next, AGE with a random mask, improving performance by $15.46\%$ over the baseline. Finally, AGE using JEPA with the learnable node sampler achieves a $26.72\%$ improvement over the baseline. Based on these experiments, we confirmed that JEPA works better than GA as expected, while node sampler further improves performance with a notable margin. Furthermore, we include an additional study on architectural design choices in the Appendix, such as reason we choose different input features for LLMs during training and inference in GraphRAG.

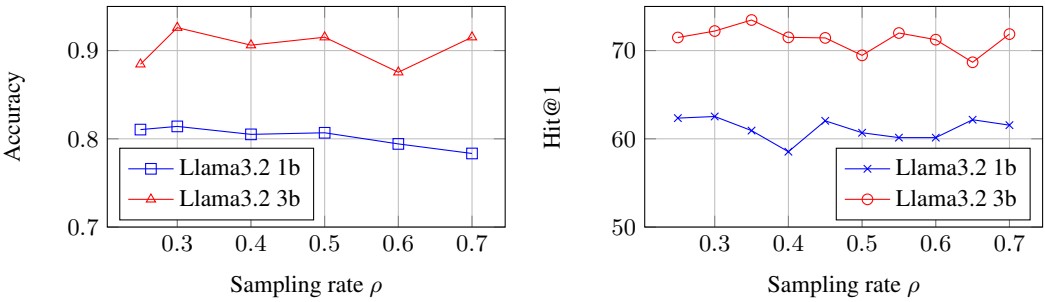

Figure 3: Performance of against sampling rate (ExplaGraphs)

Figure 4: Performance against sampling rate (WebQSP)

**Analysis on the Sampling Rate** $\rho$. Figure 3 illustrates how the sampling rate impacts AGE G-Retriever performance on ExplaGraphs and WebQSP, guiding our hyper-parameter setting. Average retrieved nodes are $18.21$ and $5.17$ on WebQSP and ExplaGraphs, respectively. A sampling rate $\rho = 0.3$ gives the best performance on ExplaGraphs with both LLM settings ($81.4\%$ for Llma3.2-1B and $92.6\%$ for Llama3.2-3B). The same setting also achieves the best performance on WebQSP for Llama3.2-1B ($62.5\%$) and the second best for Llam3.2-3B ($72.2\%$), compared to the best score of $73.5\%$ at $\rho = 0.35$. From these observations, we decided to use $\rho = 0.3$ through the experiments.

**Qualitative Evaluation.** To analyze node sampling results, we visualized the node sampling results on two samples from the ExplaGraphs test set in Figure 5. Nodes are colored based on clustered text embeddings to track node-wise feature restructuring through graph relationships. The graph

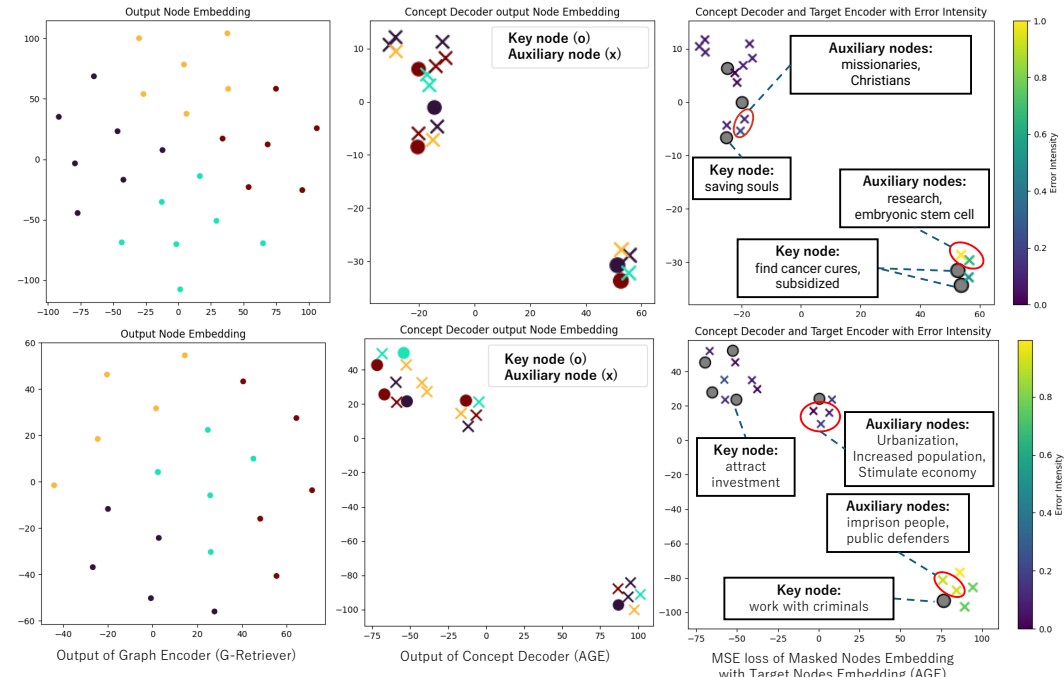

Figure 5: Node embedding of G-Retriever and AGE G-Retriever with sampling rate $\rho = 0.3$ using t-SNE van der Maaten & Hinton (2008): Nodes are colored by clustering node's text (left two), or target error (the rightest).

encoder process maintains the clustering structure of text graph embeddings in both G-Retriever (first column) and AGE (second column). In contrast, the concept encoder-decoder module (second column) shuffles the colored nodes, indicating a reorganization of the node-wise embeddings.

The last column displays text entities of some key and auxiliary nodes. Our node sampler is designed to sample entities from specific domains as key nodes. As "saving souls" is sampled as a key node, inferring "missionaries" and "Christians" from it seems easier than the reverse. We observe the same tendency with the key node "work with criminals" and the auxiliary nodes "imprison people" and "public defenders." The last column also shows the target loss for each auxiliary node using the color bar, where 1.0 represents the maximum error in the test set, and 0.0 the least. From the color visualization, we observe that non-isolated key nodes achieve lower errors in auxiliary node prediction, suggesting that relations between key nodes support the prediction. We provides additional qualitative results, including failure cases, in Appendix.

## 6 LIMITATION AND CONCLUSION

Although we achieved a significant improvement in GraphRAG, there are two minor limitations. First, we used a fixed sampling rate despite variations in key node density between the graphs. Second, we tested AGE only on GraphRAG tasks, even though it is applicable to other modalities. These limitations suggest areas for future improvement and the potential for broader applications.

We proposed Adaptive-masking for Graph Embedding (AGE) to improve structured graph embeddings and enhance LLM performance on GraphQA tasks. The method introduced JEPA, a self-supervised learning architecture which enhanced the graph-structure embedding for downstream reasoning tasks. Our node sampler demonstrated its effectiveness in the ablation study, successfully identified key nodes within given graphs. The quantitative results confirmed AGE's consistent performance gain in GraphRAG tasks while maintaining computational cost.

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
