# A  PROOF OF CONCEPT

In this section, we motivate AGE's design for representing retrieved subgraphs for higher-order reasoning skills. We assume this requires on the most common static search single-turn retrieval and LLM for academic simplicity.

**Problem definition:** Given input query $q$, we generate chain-of-thoughts response $y = i \oplus r$ (interleaved domain knowledge $i$ and reasoning steps $r$). We use a static retrieval engine that returns $T^*$ for text knowledge and $S^*$ for subgraph. The learning objective $\min_\theta \mathbb{E}[\mathcal{L}]$ optimizes representation $S^*$ to prioritize reasoning $r$ over knowledge $i$.

For the chain-of-thoughts response is defined as $y = i \oplus r$ where $y$ is the concatenation of knowledge $i$ and reasoning $r$ through three discrete generation processes below.

- **Graph Knowledge Retrieval:** Given the query $q$ on a textual graph $G$ and $S(G)$ is the set of all subgraphs of $G$. Retrieval system extracts relevant subgraph $S^* \in S(G)$ and text-modal knowledge $T^*$. Popular retrieval systems select top-k elements by cosine similarity, yielding nodes $V_k^*$ and edges $E_k^*$ considered relevant to the query. A non-optimized retriever may yield corrupted subgraphs $S^* = (V_k^*, E_k^*)$, as they may be contain redundant or lack suggestive elements.

- **Graph Knowledge Representation:** Graph embedding module is trained to represent graph that guides the LLM to produce expected answers. Embedding module $R_\omega(\cdot)$ learn to represent corrupted subgraph $S^*$ to $\overline{S^*}$ for an LLM$\Phi$ improve generation.

- **Contextualized Reasoning:** Given $q$, $T^*$ and $\overline{S^*}$, LLM synthesizes domain knowledge $t$ by recall their internal parametric knowledge with external inputs, following the conditional distribution $i \sim \pi_\theta(i|q, T^*, \overline{S^*})$. Then LLM generates reasoning steps $r$ conditioned on $q$, $T^*$, $\overline{S^*}$ with the recalled internal knowledge $i$, adhering to the reasoning distribution $r \sim \pi_\theta(r|q, T^*, \overline{S^*}, i)$.

Here we formally analyze and discuss the subgraph representation learning objectives of both vanilla embedding module and AGE embedding module with LLM generation distribution.

- Subgraph embedding module that employ GNN or Transformer:

$$\overline{S^*} = R_\omega^{\text{GNN}}(S^*) = \left\{ \overline{V}_i \right\}_{i \in V^*} \tag{1}$$

$$\overline{V}_i = \sum_{j \in \mathcal{N}(i)} \alpha_{ij} W_V V_j, \tag{2}$$

Here, $\mathcal{N}(i)$ denotes the local neighborhood of node $i$ (e.g., $\mathcal{N}(i) = \{j \mid (i,j) \in E^*\}$).

$$\overline{S^*} = R_\omega^{\text{Transformer}}(S^*) = \left\{ \overline{V}_i \right\}_{i \in V^*} \tag{3}$$

$$\overline{V}_i = \sum_{j \in V^*} \alpha_{ij} W_V V_j, \qquad \alpha_{ij} = \text{softmax}_{j \in V^*}\left( \frac{(W_Q V_i)^\top (W_K V_j)}{\sqrt{d_k}} \right), \tag{4}$$

where $W_Q, W_K, W_V$ are learnable matrices, $d_k$ is the key dimension. These formulations use attention-weighted aggregation as node embeddings. Static retrieval from graphs with high-order structural patterns (motifs/role patterns) produces corrupted subgraphs that contain redundant nodes or miss critical elements. Without explicit structural constraints, training weighted-sum aggregation through supervised or semi-supervised learning separates signal from noise, may lead to diluted node representations within the embedding space. In a frozen state, LLMs may fail to capture relationships in the embedding space due to their struggle to handle diluted node representations.

- Subgraph embedding module that employ RL-guide mask-based SSL:

$$\overline{S^*} = R\omega^{\text{AGE}}(S^*) = \left\{ \overline{V}_i \right\}_{i \in V^*} = Decoder_\phi\big( \Delta_{V_{masked}}, Encoder_\theta(V_{key}) \big) \tag{5}$$

$$= V_{key} \sim \text{Sampler}_\theta(V_{key} \mid V^*) \left[ \sum_{j \in V^*} \alpha_{ij} \left[ \Big[ \sum_{j \in V^{key}} \alpha_{ij} W_V V_j \Big]; \Delta_{V_{masked}} \right] \right] \tag{6}$$

$\Delta_{V_{masked}}$ denotes the auxiliary masked node features used as decoder input, while $V_{key}$ denotes the key node features used as encoder input and $[;]$ is the concatenation operation.

$$\begin{cases} \Delta_{V_{masked}} = Mask(V_j) & \text{if } a_j = 1 \\ V_{key} = Visible(V_j) & \text{if } a_j = 0 \end{cases} \quad \sum_{j \in V^*} \text{Sampler}_\theta(a_j \mid V_j), \qquad (7)$$

where $V_{key}$ represents key node features for encoder input, where $\text{Mask}(\cdot)$ masks features, $\text{Visible}(\cdot)$ preserves them, and $a_j \in \{0, 1\}$ denotes the binary RL action (1=mask, 0=keep) for node $j$.

By employing mask-based SSL alongside a reinforcement learning framework, AGE learns structural dependency node representations in the embedding space through reconstruction objectives. The reinforcement learning framework is used to estimate which nodes are critical for preserving graph structure and semantic information. Then, the estimated nodes are applied to guide mask-based SSL to reconstruct that provide structural constraints in the embedding space, enabling LLMs to better separate signals and capture relationships within it.

- The joint generation distribution of LLMs is:

$$\pi_\theta(y \mid q, T^*, \overline{S^*}) = \pi_\theta(i \oplus r \mid q, T^*, \overline{S^*}) = \underbrace{\pi_\theta(i \mid q, T^*, \overline{S^*})}_{\text{Knowledge Recalling}} \cdot \underbrace{\pi_\theta(r \mid q, T^*, \overline{S^*}, i)}_{\text{Contextualized Reasoning}}, \quad (8)$$

- The loss function optimizes both knowledge integration and contextualized reasoning:

$$\begin{aligned} \mathcal{L} &= -\mathbb{E}_{(q,T^*,\overline{S^*})}\big[\log \pi_\theta(i \mid q, T^*, \overline{S^*})\, \pi_\theta(r \mid q, T^*, \overline{S^*}, i)\big] \\ &= -\mathbb{E}_{(q,T^*,\overline{S^*})}\big[\log \pi_\theta(i \mid q, T^*, \overline{S^*})\big] \quad - \mathbb{E}_{(q,T^*,\overline{S^*})}\big[\log \pi_\theta(r \mid q, T^*, \overline{S^*}, i)\big]\,, \end{aligned} \quad (9)$$

Through the lens of multi-task learning, we compare above equations and from two perspectives:

- **Retrieved Subgraph Representation on LLMs' generation distribution.** Based on equation equation 8, LLMs decompose response generation into knowledge recall and contextualized reasoning. In the frozen state, diluted subgraph representations trigger a cascade—weak knowledge recall causes flawed reasoning. This limitation creates a bottleneck that degrades response quality.

- **Retrieved Subgraph Representation with Parameter-Efficient Fine-Tuning.**

  Following previous research Wang et al. (2025b), we assume the retrieved representation is **explicitly**, we have equation 9 as:

$$\mathcal{L} = \underbrace{-\mathbb{E}_{(q,T^*,\overline{S^*})}\big[\log \pi_\theta(i \mid q, T^*, \overline{S^*})\big]}_{\textbf{Loss of Integration}} \downarrow \quad \underbrace{-\mathbb{E}_{(q,T^*,\overline{S^*})}\big[\log \pi_\theta(r \mid q, T^*, \overline{S^*}, i)\big]}_{\text{Loss of Reasoning}} \uparrow,$$

$$(10)$$

The arrows indicate the loss function shifts to reasoning. That means the loss term shifts from knowledge identification to integration, that $\pi_\theta(i \mid q, T^*, \overline{S^*})$ has already reached "application" levels of retrieved graph knowledge. Therefore, explicit subgraph representation aids knowledge integration beyond mere identification during fine-tuning.

Conversely, the retrieved subgraph representation is **diluted**, we have equation 9 as:

$$\mathcal{L} = \underbrace{-\mathbb{E}_{(q,T^*,\overline{S^*})}\big[\log \pi_\theta(i \mid q, T^*, \overline{S^*})\big]}_{\textbf{Loss of Identification}} \uparrow \quad \underbrace{-\mathbb{E}_{(q,T^*,\overline{S^*})}\big[\log \pi_\theta(r \mid q, T^*, \overline{S^*}, i)\big]}_{\text{Loss of Reasoning}} \downarrow,$$

$$(11)$$

The arrows indicate that the loss function prioritizes identification. This reveals inefficient knowledge use: the model identifies patterns directly, treating retrieved subgraphs as training data. This creates a trade-off: instead of learning to apply retrieved subgraphs in reasoning, $\pi_\theta(i \mid q, T^*, \overline{S^*})$ prioritizes identifying graph structures over reasoning. Poor retrieved subgraph representation implicitly hinders reasoning capability development, forcing resources into identification tasks that better representations would cover.

## B ADDITIONAL EXPERIMENTAL DETAILS

### B.1 IMPLEMENTATION SETTINGS (AGE G-RETRIEVER)

When integrated with G-Retriever, we consistently use the AdamW Loshchilov & Hutter (2017) optimizer and set the initial learning rate at $1e - 4$, with a weight decay of 0.05. Following the baseline work He et al. (2024), we set learning rate decays with a half-cycle cosine decay after the warm-up period. To avoid overfitting, we implement early stopping with a patience of 3 epochs. The experiments used 2 NVIDIA 2080Ti-11G or 2 NVIDIA A100-80G GPUs.

**GNN.** We use Graph Transformer as the GNN backbone applied in the Graph Encoder and Graph Structure Based Aggregator. Similar to previous approaches He et al. (2024), our settings for its employ 2 layers, each with 4 attention heads, and a hidden dimension size of 1024.

**LLM.** We use the open-source Llama3.2 1B, 3B AI (2024), and Llama3.1 8B as the LLM backbone. When LoRA Hu et al. (2021) is applied with the LLM, the LoRA scaling factor hyperparameter is set to 16. Following previous work He et al. (2024), we configure the LLM with a maximum input text length of 512 and a maximum number of new tokens to generate of 32.

**Subgraph Construction.** We follow previous approaches He et al. (2024) that select the top $k$ nodes and edges through subgraph construction by setting $k$ to 3 for SceneGraphs dataset. For WebQSP dataset, $k = 3$ for nodes and $k = 5$ for edges. For the ExplaGraphs dataset, the entire graph fits within the LLM's context window. Thus, setting $k$ to 0 for retrieves the original graph without modification.

### B.2 IMPLEMENTATION SETTINGS (AGE AMAR)

When integrated with AMARXu et al. (2025), to fairly compare we keep the training settings of AMAR, setting the retrieved data to 100 on WebQSP, Soft prompt length to 7, Beam search number to 8, and Max new tokens to 256 for the WebQSP dataset. For the CWQ dataset, we set the retrieved data to 4, Soft prompt length to 16, Beam search number to 15, and Max new tokens to 256. With the Llama2 Touvron et al. (2023) is trained with LoRA learning rate $5e-5$ scaling factor hyperparameter is set to 32.

### B.3 THE CHOICE OF AGE ARCHITECTURE

#### B.3.1 THE CHOICE OF GRAPH STRUCTURE EXTRACTOR ARCHITECTURE ON AGE G-RETRIEVER

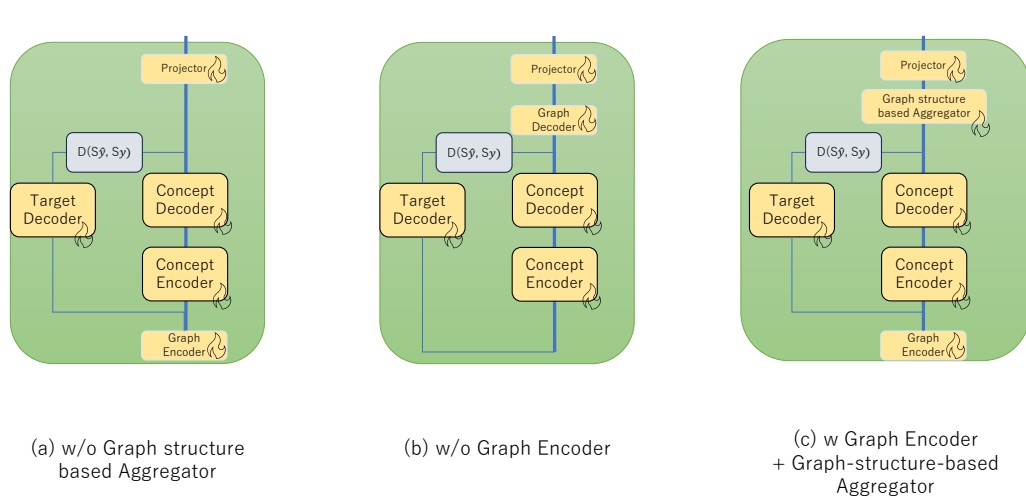

(a) w/o Graph structure based Aggregator

(b) w/o Graph Encoder

(c) w Graph Encoder + Graph-structure-based Aggregator

Figure A: Investigation of core component arrangement: We tested our JEPA LeCun & Courant (2022) architecture with three different GNN arrangements, including (a) graph encoder only, (b) graph-structure-based aggregator only, and (c) both of them.

Figure A shows the architectures with graph encoder only, graph-structure-based aggregator only, and both combined. The best-performing architecture is the combination of graph encoder and graph-structure-based aggregator. It achieves a Hit@1 score of 73.46% on the WebQSP dataset, improving upon 71.12% with Graph Encoder and 72.44% with graph-structure-based aggregator.

### B.3.2 THE CHOICE OF GNN ON AGE G-RETRIEVER

| GNNs | WebQSP | ExplaGraphs |
|------|--------|-------------|
| GCN | 56.75 | 0.8321 |
| GAT | 61.42 | 0.8212 |
| Graph Transformer | 62.53 | 0.8501 |

Table A: Performance comparison of different GNNs on Llama3.2 1B.

In Table A, our investigation extends to existing popular GNNs employed as both graph encoders and graph-structure-based Aggregator, including the Graph Convolutional Network (GCN) Kipf & Welling (2016), Graph Attention Network (GAT) Veličković et al. (2017) and Graph Transformer Shi et al. (2020).

This illustrates the performance comparison of these GNNs on the WebQSP and ExplaGraphs datasets. On the WebQSP dataset, the GCN, GAT, and Graph Transformer achieve Hit@1 scores of $56.75$, $61.42$, and $62.53$, respectively. In the ExplaGraphs dataset, the Graph Transformer achieves the highest accuracy of $0.8501$, followed by the GCN with an accuracy of $0.8321$, and the GAT trailing slightly at $0.8212$.

These findings emphasize the critical role of selecting an appropriate GNN architecture tailored to the unique properties and demands of each dataset. To maintain performance across various datasets, we choose the Graph Transformer for all experiments.

### B.3.3 THE DESIGN OF AGE WITH GENERATION ARCHITECTURE

As shown in Figure B, we provide more details of AGE with the randomly masked generative architecture (GA) introduced in Section 4.3. Designing AGE with GA aims to complement the input node to enhance the embedding of the graph-structure-based aggregator during the inference stage. To do this, we train the encoder-decoder with input nodes masked by a random mask at a masking ratio of 70%.Then, the encoder is trained to embed unmasked nodes, and the decoder reconstructs masked nodes through the target loss. On the other hand, prompt tuning loss is used to train the graph-structure-based aggregator and MLP for referring to edges $E^*$, and the MLP adjusts the aggregated embeddings to align with the LLM input dimension. During the inference stage, random masking is disabled. All input nodes are fed into the encoder-decoder to reconstruct the input for the graph-structure-based aggregator.

Table B: Analysis on the number of $GNN_{ge}$' layers with LLaMA 3.2 3B on WebQSP. GE refers to Graph Embedding.

| | PT w/o GE | G-Retriever | | AGE | | |
|---|---|---|---|---|---|---|
| # of layers | - | 2 | 4 | 1 | 2 | 4 |
| Hit@1 ($\uparrow$) | 48.3 | 64.9 | 71.3 | 73.5 | 70.5 | 69.7 |
| Training time (Min./Epoch) ($\downarrow$) | 4.5 | 4.4 | 4.5 | 4.5 | 4.6 | 4.9 |
| Inference speed (Tokens/sec) ($\uparrow$) | 88.9 | 86.0 | 84.4 | 87.6 | 84.9 | 81.1 |

### B.3.4 ANALYSIS ON THE LAYER NUMBER OF THE GRAPH ENCODER $GNN_{ge}$ ON AGE G-RETRIEVER

Compared to G-Retriever, AGE's inference path has additional modules. While this might increase processing time, Table B indicates otherwise. For G-Retrievers, a deeper $GNN_{ge}$ performs better.

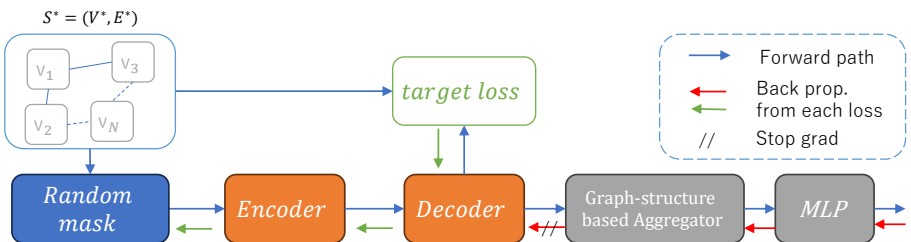

Figure B: Adaptive-masking for Graph Embedding in Generation Architecture: The node embedding module is trained on both prompt tuning loss and target loss.

In contrast, AGE performs better with fewer layers, as the added modules effectively substitute for reduced GNN layers. As a result, AGE achieves superior performance while maintaining the training time of the baseline method. We provide further analysis on computational complexity in Appendix.

### B.3.5 COMPARISON WITH DEEPER GNNs

|  | LLM | GNN Layer | Parameter | FLOPs (G) | Acc |
|---|---|---|---|---|---|
| G-Retriever | Llama 3.2 1B | 4 | 3.9 M | 0.2 G | 0.5595 |
| G-Retriever | Llama 3.2 1B | 20 | 11.3M | 1.2 G | 0.7238 |
| AGE G-Retriever | Llama 3.2 1B | 2 | 7.8 M | 1.1 G | 0.8501 |
| G-Retriever | Llama 3.2 1B | 4 | 3.9 M | 0.2 G | 0.7761 |
| G-Retriever | Llama 3.2 1B | 20 | 11.3M | 1.2 G | 0.8682 |
| AGE G-Retriever | Llama 3.2 1B | 2 | 7.8 M | 1.1 G | 0.9260 |

Table C: Compare AGE with DeeperGNN in ExplaGraphs test set

Table C compares the number of GNN layers and the performance of G-Retriever and Adaptive-masking for Graph Embedding models. G-Retriever with 20 layers is prepared as a model whose computational cost (GFLOPs) is similar to AGE with 2 layers.

Applying the G-Retriever with 20 layers largely improves performance. However, AGE still outperforms G-Retriever by approximately 10 points when using Llama 3.2 1B and 6 points when using Llama 3.2 3B, demonstrating AGE's superior performance.

### B.3.6 THE CHOICE OF CONCEPT DECODER AND NODE SAMPLER ARCHITECTURE.

Table Da illustrates our analysis of model performance across various concept decoder depths. We increased the decoder depth from 1 block to 4 blocks, thereby increasing the parameters from 81M to 106M. Despite this increase, the performance decreased, with scores dropping from 62.5 to 57.4 on WebQSP. The best performance is achieved with a decoder depth of 1 on both ExplaGraphs and WebQSP. Therefore, we choose a single transformer block to maintain the performance of the concept decoder in this work.

As shown in Table Db, we investigate different network architectures for the node sampler design. Increasing the number of transformer blocks leads to marginal gains in performance on the WebQSP dataset, although it requires more memory. To maintain computational and performance efficiency, we selected a single transformer block with a hidden dimension of 1024 for the node sampler in all subsequent experiments.

### B.3.7 THE STUDY ON KEY NODES SAMPLING

Figure C and D illustrates how the static strategy node sampler and RL-based node sampler performance in various sampling rate on ExplaGraphs and WebQSP, guiding our node sampler architecture setting.

| Decoder Depth | Parameter. (M) | ExplaGraphs (Acc) | WebQSP (Hit@1) |
|:---:|:---:|:---:|:---:|
| 1 | 81 | 0.8501 | 62.5 |
| 2 | 85 | 0.7978 | 61.2 |
| 4 | 106 | 0.8123 | 57.4 |

(a) Concept Decoder

| Depth | $d$ | FLOPs (G) | ExplaGraphs (Acc) | WebQSP (Hit@1) |
|:---:|:---:|:---:|:---:|:---:|
| 1 | 1024 | 1.1 | 0.8501 | 62.5 |
| 1 | 2048 | 1.6 | 0.8213 | 59.3 |
| 2 | 1024 | 1.4 | 0.7906 | 63.1 |

(b) Node Sampler

Table D: Ablation studies for network architecture design.

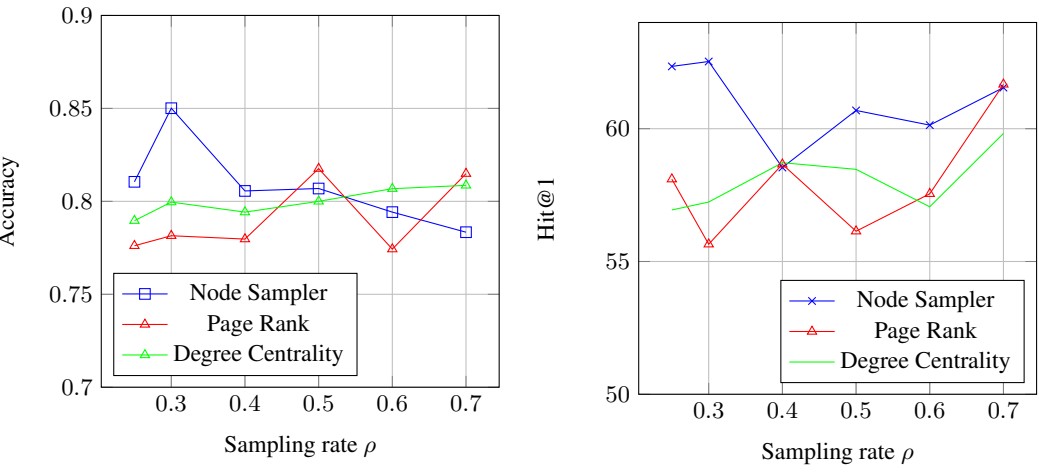

Figure C: Relationship of sampling rate with key node sampling strategy and performance (on ExplaGraphs wit Llama 3.2 1B)

Figure D: Relationship of sampling rate and performance (on WebQSP)

When using static PageRank Page et al. (1998) and Degree Centrality strategies for node sampling, higher sampling rates tend to better performance. However, this suggests that the key nodes that identified by these static methods are not sufficiently impactful. Lead to the Concept Encoder-Decoder needs a larger set of key nodes to effectively embed the graph, which then helps guide the LLM to produce the desired answers.

In contrast, RL-based node samplers can achieve high performance with lower sampling rates. This indicates that the key nodes chosen by RL-based methods more effectively support the Concept Encoder-Decoder, boosting the quality of the graph embedding. As a result, the LLM can produce the expected answers with fewer key nodes involved in the guidance process.

### B.3.8 THE STUDY ON TRANSFERABILITY

| Method | LLM | WebQSP → ExplaGraph | ExplaGraph → WebQSP |
|:---|:---|:---:|:---:|
| G-Retriever | Llama 3.2 1B | 0.5106 | 36.48 |
| **AGE G-Retriever** | Llama 3.2 1B | 0.5685 | 39.25 |
| G-Retriever | Llama 3.2 3B | 0.4404 | 50.35 |
| **AGE G-Retriever** | Llama 3.2 3B | 0.6021 | 53.53 |

Table E: Cross-Dataset Transfer Learning Performance.

Table E show the transferability of AGE when interacted with G-Retriever. AGE support G-Retriever to strong transferability to transfer learned graph embedding encoding capabilities across datasets. When trained on a large dataset, AGE can enhance generation on a smaller dataset using the trained model. Notably, AGE trained on WebQSP on ExplaGraphs with Llama 3.2 3B outperforms transferability of GRAG.

### B.3.9  THE STUDY ON NUMBER OF RETRIEVAL

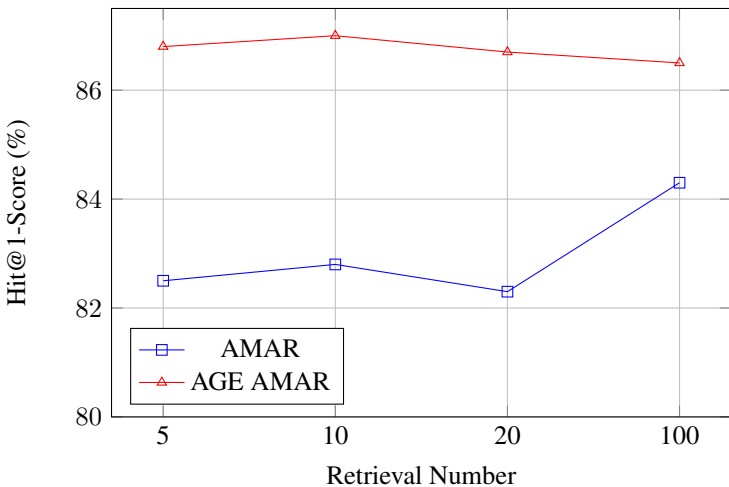

Figure E: Performance impart on vary number of retrieval on WebQSP

As illustrated in FigureE, AMAR was designed to address the challenge posed by excessively long retrieved data inputs and to leverage rich information more effectively. However, when the volume of retrieved data is relatively small, AMAR's performance shows minimal improvement, indicating that the recalled information is insufficient. In such cases, AGE is capable of mapping retrieved data to useful embeddings, eading to significant performance improvements, with scores of 86.8 for 5 retrievals and 87.0 for 10 retrievals.

Conversely, when large amounts of data are retrieved, the accompanying noise complicates the ability of LLMs to identify and prioritize the most relevant information. AGE consistently maintains its performance with minimal variation, highlighting the robustness as 86.5. Moreover,to fairly compare with AMAR, we choose 100 retrievals with an Hit1@ of 86.5.

### B.3.10  IMPART OF AGE WITH LoRA FINETUNING PERFORMANCE

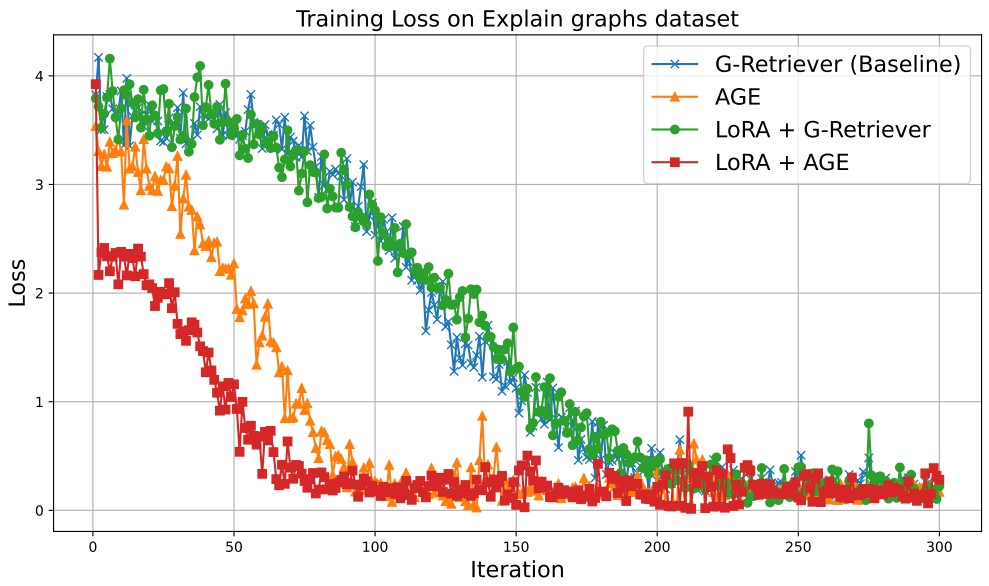

Figure F: Training loss on Explain Graphs.

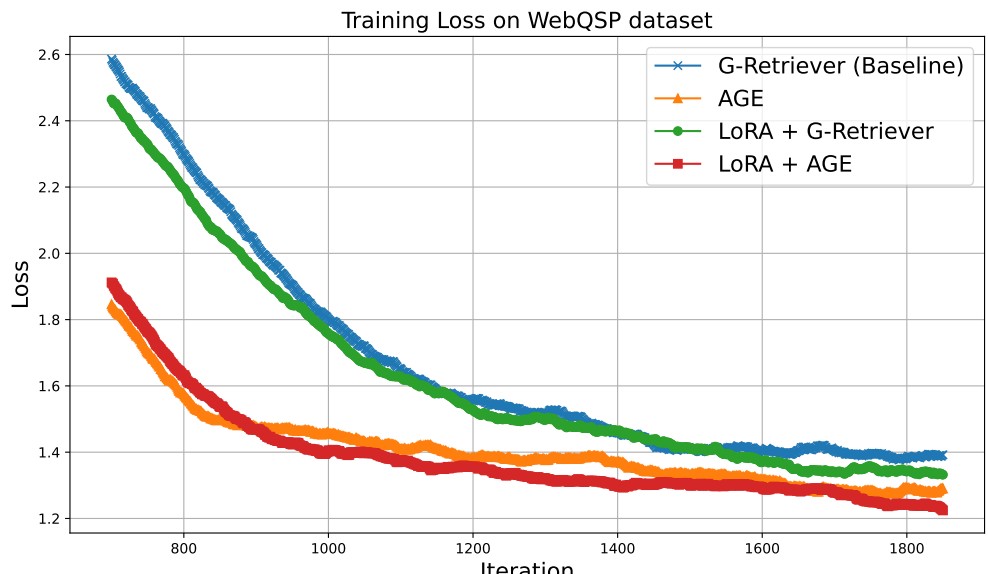

Figure G: Training loss on WebQSP

Table F: The study of sampling strategy on Node Sampler (with Llama3.2 1b on PEFT, on Expla-Graphs and WebQSP).

|  | Gumbel-Softmax | Straight-Through (ST) Estimator | REINFROCE |
|---|---|---|---|
| ExplaGraphs | 0.8378 | 0.8241 | 0.8501 |
| WebQSP | 68.5 | 66.7 | 69.1 |

We conduct extensive experiments to compare the trend of training loss with G-Retriever baselines, as illustrated in FigureF and G. AGE provides structural constraints in the embedding space through mask-based SSL with reconstruction objective, enabling LLMs to better separate signals and capture relationships within it, leading to increased convergence rate and lower loss observed in the initial training stage compared with the G-Retriever.

### B.3.11  THE CHOICE OF RL METHOD FOR NODE SAMPLER

Table F show Both Gumbel-Softmax and REINFORCE demonstrate strong performance on the ExplaGraphs dataset, outperforming the Straight-Through Estimator. On the WebQSP dataset, RE-INFORCE leads slightly, indicating it may be the most effective method among the three tested for this task. From these observations, we decided to use REINFROCE through the experiments.

### B.3.12  THE CHOSE DIFFERENT INPUT FEATURES FOR LLMS DURING TRAINING AND INFERENCE

Table G show AGE with $h_{out}$ as LLMs input in inference state outperforms AGE with $h_{target}$ as input on both datasets. This indicates that representing nodes using a concept encoder-decoder

Table G: Performance of AGE in PEFT (with Llama3.2 1b , on ExplaGraphs and WebQSP).

|  | PEFT G-Retriever | AGE w $h_{target}$ as LLM input | AGE w $h_{out}$ as LLM input |
|---|---|---|---|
| ExplaGraphs | 0.7328 | 0.8212 | 0.8501 |
| WebQSP | 65.3 | 66.7 | 69.1 |

Table H: Performance of Target Encoder on ExplaGraphs, trained with PEFT using Llama3.2 1b.

| Loss type | w/o norm + w/o EMA | norm | EMA | norm + EMA |
|---|---|---|---|---|
| L1 | 0.7978 | 0.8375 | 0.8194 | 0.8303 |
| MSE | 0.8357 | 0.8501 | 0.8375 | 0.8501 |

$h_{out}$ is more effective than the target encoder in downstream LLMs input tasks. Based on these observations, we concluded that designing the connection of $h_{target}$ during training and $h_{out}$ during inference to the downstream LLM not internalizes the learning-inference mismatch. Instead, it allows the student model has already surpassed the performance of the teacher model, allowing for a more robust representation.

### B.3.13 THE STABILITY OF TARGET ENCODER

Due to differences in node representations during the training inference stage, using identical parameters for both the concept encoder and the target encoder helps prevent distributional shifts. We apply two popular techniques to enhance the stability provided by the target encoder for the concept encoder-decoder during the training stage. EMA weights are defined as an exponential moving average of the encoder weights, and normalization is applied to enhance stability during the learning process. Normalization ensures consistent activation distributions and reduces internal covariate shift. Table H shows that MSE performs better with L1, and normalization alone achieved the highest score. These results indicate that normalization consistently improves encoder stability and performance, and adding EMA offers further enhancements.

### B.3.14 THE LANDSCAPE OF EXISTING KGQA METHODS

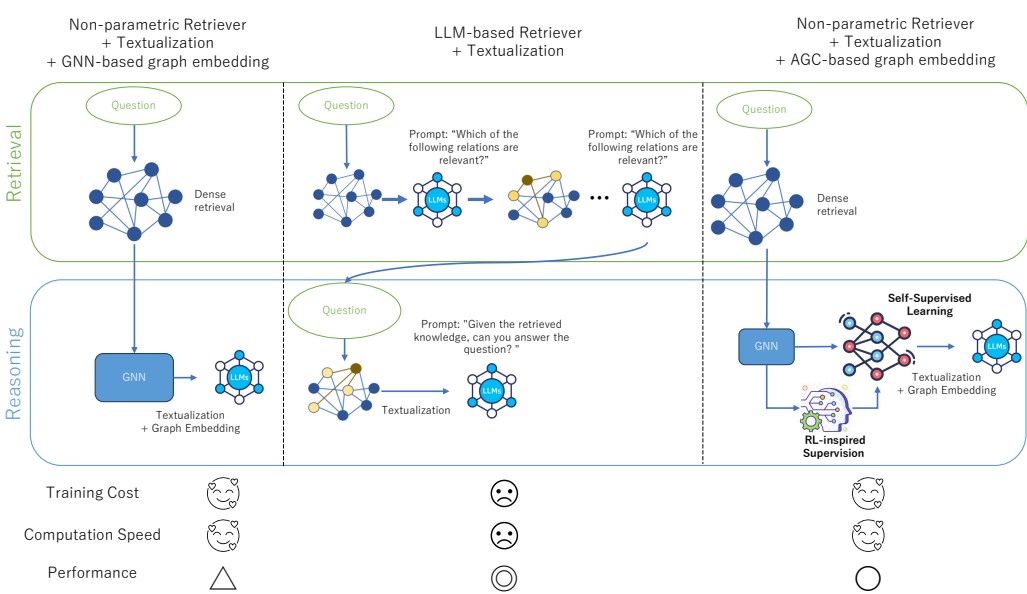

Figure H: The landscape of existing KGQA methods. GNN-based methods reason on dense subgraphs as they can handle complex and graph information. LLM-based methods employ the same LLM for both retrieval and reasoning due to its ability to understand natural language.

Figure H illustrates the spectrum of current Knowledge Graph Question Answering (KGQA) approaches regarding KG retrieval and reasoning capabilities. Graph Neural Network (GNN)-based methods, including NSM He et al. (2021), ReaRev Mavromatis & Karypis (2022), and G-Retriever He et al. (2024), perform reasoning on retrieved dense subgraphs by utilizing the GNN to embed graph structures.

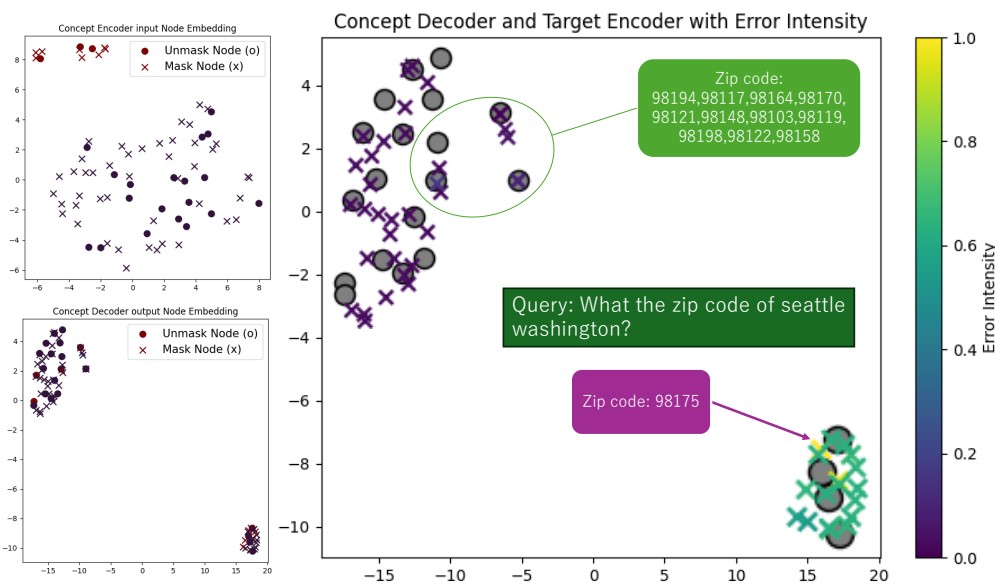

Figure I: Failure case visualization of AGE on WebQSP dataset.

Recent LLM-based methods leverage the power of LLMs for both retrieval and reasoning. ToG Sun et al. (2024) uses the LLM to retrieve relevant facts hop-by-hop. RoG Luo et al. (2024) uses the LLM to generate plausible relation paths which are then mapped on the KG to retrieve the relevant information. However, the frequent calls to the LLM significantly increase the training and inference costs.

In this work, we improve LLM reasoning by enhancing the graph embedding of the GNN method with RL-inspired supervision integrated into the SSL framework. This improves the performance of the non-parametric retriever to levels comparable to those of LLM-based retrievers.

## B.4  ADDITIONAL QUALITATIVE EVALUATION

We provide additional visualizations in Figures J, K on the ExplaGraphss dataset and Figure L on WebQSP dataset.

In the first row of Figure J, we consider the addition of node text information and its visualization. It is easier to infer the auxiliary node embeddings "Payday loans" and "For the disadvantaged" from an key node embeddings "Provide assistance". Conversely, it is more challenging to infer the key node embeddings "Provide assistance" from the auxiliary node embeddings "help society" and "available".

Similarly, it is easier to infer "Bullying, However they like, Banned" mask node embeddings from a "Expensive clothes, Students" key node embeddings. Conversely, it is more challenging to infer a "Expensive clothes, Students" mask node embeddings from a "Bullying, However they like, Banned" key node embeddings.

In the second row of Figure J, when considering the addition of node text information and its visualization, it is easier to infer the auxiliary node embeddings "Motivation" and "Students work harder" from the key node embedding "Student loans". Conversely, it is more challenging to infer a "Student loans" auxiliary node embeddings from "Motivation" and "Students work harder" key node embeddings. Similar things are shown in Figure K and Figure L. **Failure Case Analysis:** Furthermore, we provide a failure case on the WebQSP dataset where AGE was trained with LLaMA 3.2 1B using a sampling rate of 0.3. In this case, the query is "What is the zip code of Seattle, Washington?". Based on the query, the retriever is provided with the node "98175," which is one of the true answers. However, the response of the LLM lacks this node. To analyze this, we visualized the node sampling results from the concept encoder's input and the concept decoder's output, as shown in

| Method | LLM | Hit@1 | Training Time (min/epoch) |
|--------|-----|-------|---------------------------|
| G-Retriever | Llama 2 7B | 70.5 | 6.2 |
| AGE G-Retriever | Llama 3.2 1B | 62.5 | 2.0 |
| AGE G-Retriever | Llama 3.2 3B | 73.5 | 4.5 |
| AGE G-Retriever | Llama 3.2 8B | 78.3 | 6.4 |
| G-Retriever | Llama 2 7B LoRA | 73.8 | 6.9 |
| AGE G-Retriever | Llama 3.2 1B LoRA | 69.1 | 2.4 |
| AGE G-Retriever | Llama 3.2 3B LoRA | 77.3 | 5.9 |
| AGE G-Retriever | Llama 3.2 8B LoRA | 80.3 | 7.3 |
| AGE AMAR | Llama 2 7B LoRA | 86.5 | 8.7 |

Table I: Training cost of AGE G-Retriever on the WebQSP dataset.

| | G-Retriver | AGE | AGE | AGE |
|---|---|---|---|---|
| LLM size | Llama 2 7b | Llama 3.1 8b | Llama 3.2 3b | Llama 3.2 1b |
| All Parameters (B) | 6.8 | 8.1 | 3.3 | 1.3 |
| Trainable Parameters (B) | 0.041 | 0.087 | 0.078 | 0.072 |
| Inference speed (Tokens/sec.) | 97.0 | 81.4 | 87.6 | 148.5 |
| Hit@1 | 70.49 | 78.25 | 73.46 | 62.53 |

Table J: Inference speed of AGE on the WebQSP dataset.

Figure I. In the first column (top row), similar to the above visualization, the graph encoder process maintains the clustering structure of text graph embeddings to provide input to the concept encoder. Additionally, the output of the concept encoder-decoder module (bottom row) shuffles the colored nodes. In the second column, the target loss of each auxiliary node is represented using a color bar, where 1.0 indicates the maximum error in the test set and 0.0 the minimum error. In this column, the top left side nodes "98194, 98117, 98164,..." include low target loss auxiliary nodes, and the key nodes are the true answers. In parallel, the bottom right side node "98175" is an auxiliary node with a high target loss. This may be why the LLM omits this node in its response, and adjusting the trainable sampling rate could be a solution.

## B.5 DISCUSSION ON THE COMPLEXITY

### B.5.1 TRAINING COMPUTATIONAL RESOURCES ON AGE G-RETRIEVER

Following the previous G-Retriever He et al. (2024) method, we utilized the same two A100 GPUs, each with 80GB of memory, and conducted tests on the Llama3-8b, Llama3.1-1B, and Llama3.1-3B on WebQSP datasets. Our experiments had a training batch size of 16 and an evaluation batch size of 32, yielding the following results in Table I for training cost and Table J for validation speed.

The Table I shows the training speed and performance of AGE on the WebQSP dataset. The PEFT setting, without the graph RAG component, takes 18.7 min/epoch through prompt tuning and 19.0 min/epoch when applied with LoRA. Subsequently, the G-Retriever approach via graph RAG reduces graph size and speeds up training time.

By enhancing the embedding module on the graph RAG component, AGE with Llama3.1 8B achieves a higher Hit@1 of 78.25 in 6.4 minutes per epoch. In the tuned LLM setting, AGE with Llama3.1 8B and LoRA achieves a Hit@1 of 80.34 in 7.3 minutes per epoch. These results highlight that AGE with Llama3.2 3B outperforms G-Retriever with Llama2 7B, achieving better performance without longer training time.

| | LLM | Non-parameter Retriever | Trainable Retriever | | WebQSP | CWQ |
|---|---|---|---|---|---|---|
| | | | GNN | LLM | Hit@1 | Hit@1 |
| ToG Sun et al. (2024) | Llama2-70B | | | ✓ | 68.9 | 57.6 |
| RoG Luo et al. (2024) | Llama2-7B | | | ✓ | 74.2 | 56.4 |
| ReKnoS Wang et al. (2025a) | Llama3.1-8B | | ✓ | ✓ | 67.9 | 56.7 |
| DualR Liu et al. (2024) | Llama2-13B | | ✓ | ✓ | 78.3 | 58.0 |
| StructGPTJiang et al. (2023) | ChatGPT | | | ✓ | 72.6 | 55.3 |
| ToG Sun et al. (2024) | ChatGPT | | | ✓ | - | 76.2 |
| ToG-2 Ma et al. (2024) | ChatGPT | | | ✓ | 81.1 | - |
| RoG Luo et al. (2024) | ChatGPT | | | ✓ | - | 80.0 |
| ReKnoS Wang et al. (2025a) | ChatGPT | | ✓ | ✓ | 81.1 | 58.5 |
| GNN-RAG Mavromatis & Karypis (2024) | ChatGPT | | ✓ | | 85.7 | 66.8 |
| PoG Chen et al. (2024) | ChatGPT | | | ✓ | - | 82.0 |
| DualR Liu et al. (2024) | ChatGPT | | ✓ | ✓ | - | 82.8 |
| KBQA Xiong et al. (2024) | GPT-4 | | | ✓ | 72.5 | - |
| ReKnoS Wang et al. (2025a) | GPT-4 | | ✓ | ✓ | 84.9 | 68.2 |
| ToG Sun et al. (2024) | GPT-4 | | | ✓ | 82.6 | 69.5 |
| PoG Chen et al. (2024) | GPT-4 | | | ✓ | 87.3 | 75.0 |
| DualR Liu et al. (2024) | GPT-4 | | ✓ | ✓ | 87.6 | 73.6 |
| GraphToken Perozzi et al. (2024) | Llama2-7B | ✓ | | | 57.1 | - |
| G-Retriever He et al. (2024) | Llama2-7B-LoRA | ✓ | | | 70.2 | - |
| AMAR Xu et al. (2025) | Llama2-7B-LoRA | ✓ | | | 84.3 | 82.9 |
| AMAR Xu et al. (2025) | Llama2-13B-LoRA | ✓ | | | 83.3 | 83.1 |
| **AGE G-Retriever** | Llama3.1 8B-LoRA | ✓ | | | 80.3 | - |
| **AGE AMAR** | Llama2-7B-LoRA | ✓ | | | 86.5 | 85.2 |
| **AGE AMAR** | Llama2-13B-LoRA | ✓ | | | 86.2 | 85.1 |

Table K: Performance comparison of trainable retriever with AGE.

### B.5.2 INFERENCE COMPUTATIONAL RESOURCES ON AGE G-RETRIEVER

Table J presents the validation speed and performance of various AGE configurations on the We-bQSP dataset. Among the AGE models, Llama 3.2 3B model offers a balanced performance with a Hit@1 and an inference speed of 87.6 tokens per second. The AGE with Llama 3.2 1B achieves a significantly higher inference speed of 148.5 tokens per second while maintaining a lower Hit@1. This increased speed can be attributed to the reduced number of parameters in the 1B model, which allows for faster computation and more efficient processing, albeit at the expense of some accuracy.

These results indicate that while higher parameter models like AGE+Llama 3.1 8B provide superior accuracy, lower parameter models such as AGE+Llama 3.2 1B offer significantly increased processing speeds, supporting diverse application requirements.

### B.5.3 COMPARISON WITH TRAINABLE RETRIEVER METHODS

AGE, utilizing a non-parametric retriever, achieves accuracy levels comparable to state-of-the-art models that employ trainable parametric retrievers. As shown in Table K, AGE (Llama3.1 8B-LoRA with a non-parametric retriever) attains a Hit@1 score of 80.3% on the WebQSP dataset, closely approaching DualR (ChatGPT with a parametric retriever), which achieves 82.8%. This demonstrates that AGE effectively bridges the performance gap between non-parametric and para-metric retriever models, achieving high accuracy without the additional complexity and training overhead associated with parametric retrievers. This performance notably surpasses other models employing non-parametric retrievers, such as GraphToken (Llama2-7B) with 57.1% and G-Retriever (Llama2-7B-LoRA) with 70.2%.

The substantial increase in accuracy demonstrates that AGE enhances reasoning capabilities without relying on trainable parametric retrievers. This positions AGE as a leading approach within non-parametric retriever frameworks, closing the performance gap with models that utilize more complex and resource-intensive trainable retrievers. AGE can be deployed to train and perform inference on two RTX 2080Ti 11GB GPUs or one A100 80GB GPU.

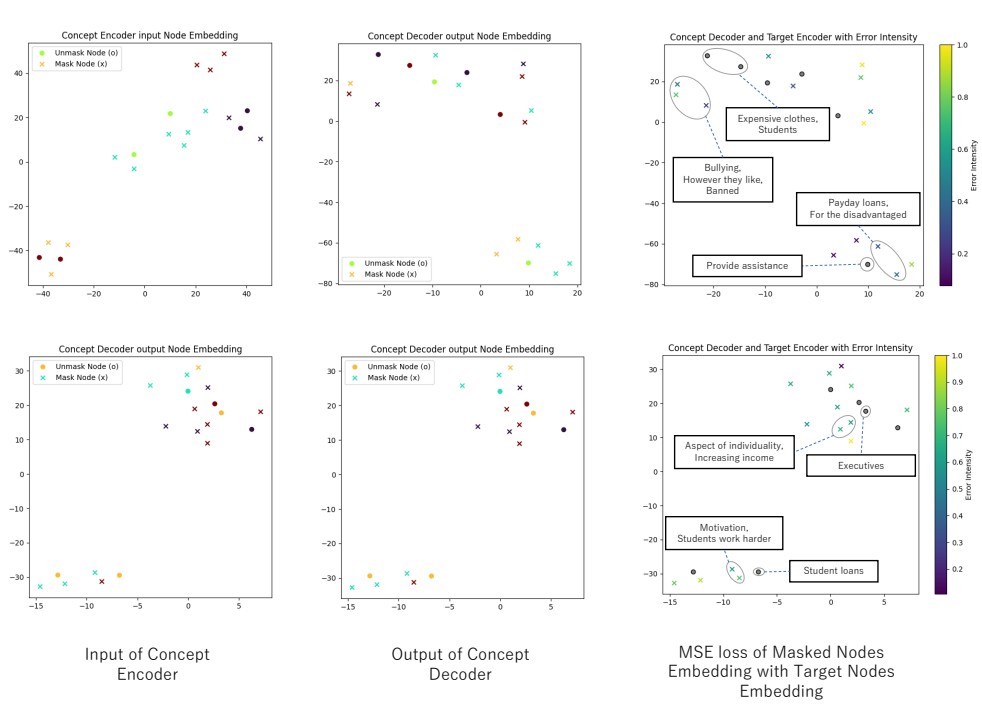

Figure J: An example visualization of AGE on the ExplaGraphs dataset.

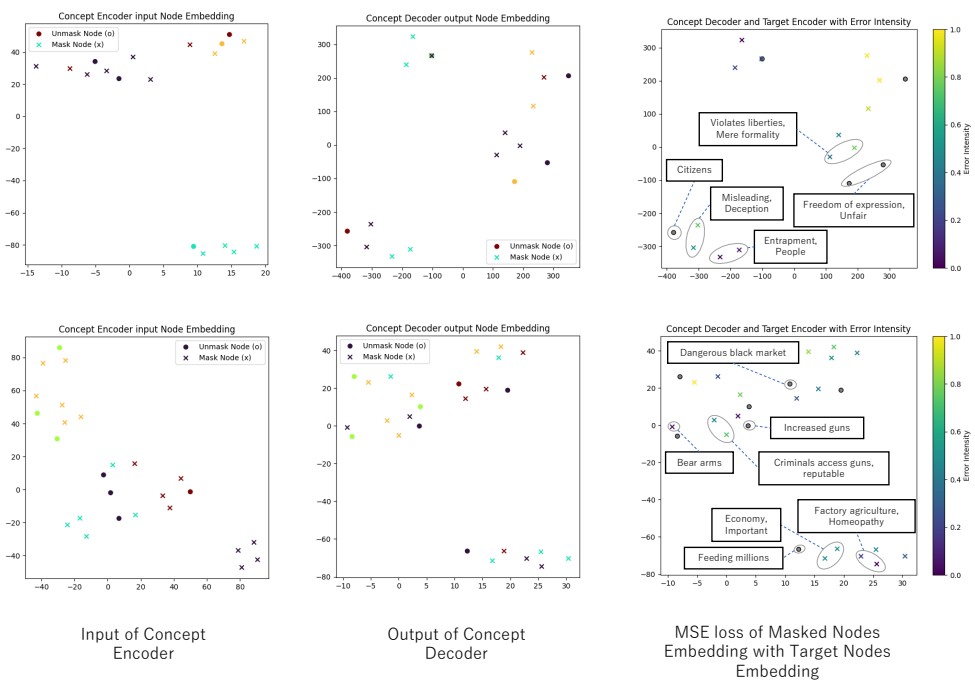

Figure K: Another example visualization of AGE on the ExplaGraphs dataset.

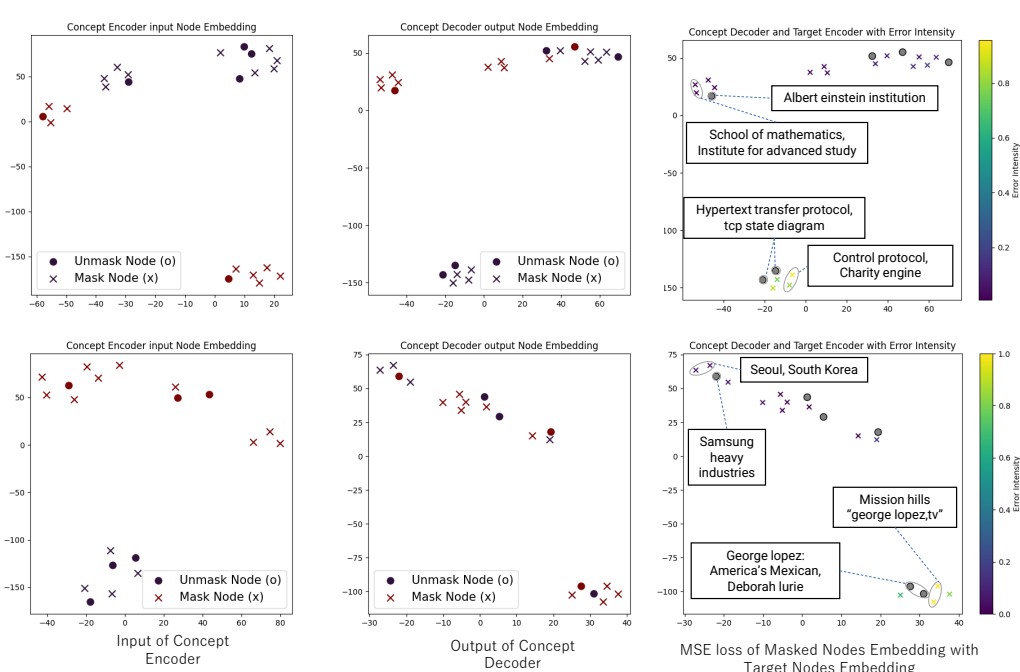

Figure L: An example visualization of AGE on the WebQSP dataset.