# OpenReview forum: "AGE: Adaptive-masking for Graph Embedding in Graph Retrieval-Augmented Generation"
_ICLR.cc/2026/Conference — Submitted to ICLR 2026_

### Official Review · Reviewer_J96N · 2025-10-28

[review text omitted: it was posted to a different submission]

---

> ### Author Response · Authors · 2025-11-15
>
> Thank you for the valuable constructive feedback. Below we respond to your questions and concerns.
>
> __Q1:__ What exactly is the GNN learning from the reasoning traces — structure, text semantics, or something else?
>
> __Response:__ To clarify what the GNN is actually learning from the reasoning traces, as well as the motivation behind architecture design and applying distributed loss on each module. We analyze the learning objective for expected $\overline{S^{*}}$ representation for LLM $\pi_{\theta}$ on LoRA finetuning as (__see Supplementary Material A for mathematical notation__):
>
> \begin{aligned}
> \mathcal{L} = \\underbrace{ -\\mathbb{E}\_{ (S^*) } \left[ \log R{\omega} (\overline{S^\*} | S^\* ) \right]  }\_{\\text{Loss of Graph Representation Module}}
> \\; \\;
> \\underbrace{-\\mathbb{E}\_{(q,T^{\*}, \overline{S^\*})}
> \left[\log \pi_{\theta}(i \mid q, T^{\*},\overline{S^{\*}})\,  \pi_{\theta}(r \mid q, T^{\*},\overline{S^\*}, i)\right]
> }\_{\\text{Loss of LLM}}
> \end{aligned}
>
> According to Bayes' Theorem, given an input $X$, a target $Y$, and latent rationales $Z$, we can sample these latent rationales $Z$ from the posterior distribution $P(Z|X, Y)$. This posterior represents the probability of latent $Z$ given both the input $X$ and the target $Y$. To compute the marginal likelihood of obtaining answer $Y$ given input $X$, we marginalize over all possible rationales $Z$:
>
> \begin{align}
> P(Y|X)
> &= \sum_{Z \sim P(Z|X,Y)} P(Z, Y|X)  \\; \\; (a) \\\\
> &= \sum_{Z \sim P(Z|X,Y)} P(Z|X) \cdot P(Y|X, Z) \\; \\; (b)
> \end{align}
>
> The equations above show how to compute the marginal likelihood $P(Y|X)$. Equation (a) makes explicit that $Z$ is sampled from the posterior distribution $P(Z|X,Y)$. Equation (b) applies the chain rule of probability to decompose $P(Z, Y|X)$ into two components: $P(Z|X)$ and $P(Y|X,Z)$.
> Following this analysis, we apply it to the learning objective for target representation $\overline{S^{*}}$ given input $S^{\*}$, latent $\mathcal{Z}$ from a posterior $R{\omega}(\mathcal{Z}|S^{\*}, \overline{S^{\*}})$ that bridges $S^{\*}$ and $\overline{S^{\*}}$. The marginal likelihood of $\overline{S^{\*}}$ given $S^{\*}$ is:
>
>
>
> \begin{aligned}
> R\omega(\overline{S^\*}|S^\*)
> &=\sum_{Z\sim R\omega(Z|S^\*,\overline{S^\*})}R\omega(Z,\overline{S^\*}|S^\*) \\\\
> &=\sum_{Z\sim R\omega(Z|S^\*,\overline{S^\*})}R\omega(Z|S^\*)\cdot R\omega(\overline{S^\*}|S^\*,Z)
> \end{aligned}
>
> Above analysis shows that learning objective Graph Representation implicitly learns to identify the latent $Z$ and map it to the expected $\overline{S^{*}}$ for LLM. The extension of the loss function is:
> \begin{aligned}
> -\mathbb{E}\\!\left[\log_{R{\omega}}(\overline{S^{\*}} \mid S^{\*})  \right]
> \&=
> \\underbrace{
> -\mathbb{E}%_{(S^{\*},\overline{S^{\*}})}
> \\! \\left[ \log R{\omega}(\mathcal{Z} \mid S^{\*},\overline{S^{\*}}) \\right]
> }\_{\\text{Loss of Latent Identification}}
> \\; \\;
> \\underbrace{
> -\mathbb{E}\\!\left[
> \log R{\omega}(\mathcal{Z} \mid S^{\*}) \cdot R{\omega}(\overline{S^{\*}} \mid S^{\*},\mathcal{Z}))
> \right]
> }\_{\\text{Loss of Representation}}
> \end{aligned}
>
> Instead of using a single model for both latent $\mathcal{Z}$ identification and $\overline{S^{*}}$ representation learning. We explicitly separate the learning into $Sampler_{\theta}$ for latent identification and $Encoder_{\theta}$-$Decoder_{\theta}$ for representation as:
>
> \begin{aligned}
> -\mathbb{E}\\!\left[\log_{R{\omega}}(\overline{S^{\*}} \mid S^{\*})  \right]
> &\approx
> \\underbrace{
> -\mathbb{E}\_{(S^{\*}, \overline{S^{\*}})} \\!
> \left[
> V\_{\\text{key}} \\in \\mathcal{Z} \sim \log \\text{Sampler}\_\theta(\mathcal{Z} \mid S^{\*},\overline{S^{\*}})
> \right]
> }\_{\\text{Loss of Node Sampling}} \\\\ \&
> \\underbrace{
> -\mathbb{E}\_{ ( ( \text{V}\_{key} \\; \, \\;  \Delta\_{V\_{masked}}) \\in S^{\*} )}
> \\! \left[
> \log \\text{Encoder}\_{\theta} (\mathcal{Z} \mid \text{V}\_{key}) \cdot \\text{Decoder}\_{\theta}  (\overline{S^{\*}} \\mid \Delta\_{V\_{masked}},\mathcal{Z})
> \right]
> }\_{\\text{Loss of Encoder-Decoder}}
> \end{aligned}
>
> By separating the learning processes, the target encoder learns the representation directly, while the encoder-decoder learns to reconstruct this representation through __Evidence Lower Bound (ELBO)__ optimization. Specifically, the node sampler learns to extrapolate $V_{key} \in \mathcal{Z}$, rendering static sampling illogical. Therefore, our node sampler with encoder-decoder architecture and explicit loss distribution yields efficient learning signals, faster convergence, and improved graph representations.
>
> To support our analysis, we provide empirical comparison experiments on sampling strategies in Section B.3.7 of the Supplementary Material, and an evaluation of the stability of the target encoder (teacher module) for the driven encoder-decoder in Section C.3.13 of the Supplementary Material.

---

> ### Author Response · Authors · 2025-11-15
>
> __Q2:__ Can you show any qualitative examples where the distilled GNN performs a reasoning-like behavior the baseline cannot?
>
> __Response:__ As Supplementary Material Figure I (right side) of Section B.4 Additional Qualitative Evaluation, which illustrates Concept Decoder and Target Encoder Error Intensity, show that Concept Decoder splits the nodes into two distinct clusters: one in the upper-left region and the other in the bottom-right region. The upper-left cluster, containing nodes such as “98194, 98117, 98164, …,” represents auxiliary nodes with low target loss and the key nodes  which correspond to the true answers. In contrast, the bottom-right cluster consists of nodes with high target loss, which are mostly incorrect answers. This demonstrates that the distilled model exhibits reasoning-like behavior that the baseline model cannot achieve.
>
>
> __Q3:__ How robust is the approach to noisy or low-quality reasoning outputs?
>
> __Response:__ We appreciate the reviewer’s insightful observation. __Continuing from Question 1__ , let us define $r$ as the reasoning trajectory, the generation distribution of LLMs with graph representation model $R{\omega}$ is (__see Supplementary Material A for mathematical notation__):
>
> \begin{aligned}
> \mathcal{L}=
> \underbrace{
> -\mathbb{E}
> \\!  \\left[ \log R{\omega}(\mathcal{Z} \mid S^{\*},\overline{S^{\*}}) \right]
> }\_{\text{Loss of Latent Identification}}
> \\; \\;
> \\underbrace{-\mathbb{E} \\!
> \\left[
> \log R{\omega}(\mathcal{Z} \mid S^{\*}) \cdot R{\omega}(\overline{S^{\*}} \mid S^{\*},\mathcal{Z}))
> \right]
> }\_{\text{Loss of Representation}}
> \\\\
> \\ \underbrace{\underbrace{
> -\mathbb{E} \\!
> \\left[
> \log \pi\_{\theta}(i \mid q, T^{\*},\overline{S^{\*}})
> \right]
> }\_{\text{Loss of Knowledge Recalling}}
> \\; \\;
> \\underbrace{
> -\mathbb{E}\\!
> \left[
> \log \pi_{\theta}(r \mid q, T^{\*},\overline{S^{\*}}, i)
> \right]
> }\_{\text{Loss of Contextualized Reasoning}}
> }\_{\text{Frozen}}
> \end{aligned}
>
> We observe that $R_\omega$ explicitly learns to identify the latent $\mathcal{Z}$ from $S^\*$ for LLM-expected $\overline{S^\*}$.
> During training with frozen LLM parameters, $R_\omega$ implicitly captures latent identification $i$ by satisfying the frozen LLM's expectations: $ Retriever(S^* \mid q, T^\*) R_\omega(\overline{S^\*} \mid S^*, \mathcal{Z}) \approx \pi_\theta(\overline{S^\*} \mid q, T^\*, i)$, yielding $\mathcal{Z} \subseteq i$. Therefore, $R_\omega$ able to learns a subspace of the frozen LLM's complete latent space through objective.
> Throughout this analysis, we argue that leveraging a learned latent space $\mathcal{Z}$, robustly restructured into the LLM-expected representation $\overline{S^\*}$, can directly improve knowledge recall and indirectly enhance reasoning.

---

> > ### Author Response · Authors · 2025-11-28
> >
> > Dear Reviewer J96N,
> >
> > We hope this message finds you well.
> >
> > As the discussion period ends is approaching, we would greatly appreciate it if you could kindly acknowledge and respond to our rebuttal for Submission 6889. Thank you again for your comments and suggestions to improve our paper, and we look forward to your reply.
> >
> > Best regards, Authors of Submission 6889

---

### Official Review · Reviewer_r4Hw · 2025-10-30

**Soundness:** 2
**Presentation:** 2
**Contribution:** 2
**Rating:** 4
**Confidence:** 4

**Summary:**

The paper addresses the issue of misaligned graph representations for frozen LLMs. It proposes Adaptive-masking for Graph Embedding (AGE), a framework that employs a mask-based self-supervised learning (SSL) process incorporating the Joint-Embedding Predictive Architecture (JEPA) and introducing a reinforcement learning–trained node sampler. AGE demonstrates its effectiveness on graph QA tasks, including ExplaGraphs, SceneGraphs, WebQSP, and CWQ. The main contributions are: (i) proposing a new graph representation framework, AGE; (ii) replacing random masking with a node sampler to distinguish key nodes from auxiliary nodes; and (iii) conducting benchmark experiments, achieving state-of-the-art accuracy or Hit@1 results on three datasets.

**Strengths:**

1. The AGE framework addresses the problem of graph representation in LLMs by leveraging the effective SSL paradigm JEPA and introducing a new node sampler to replace random masking, thereby improving performance.

2. The experimental results overall demonstrate the effectiveness of the proposed framework across four benchmarks, achieving SOTA performance on three of them.

3. The structure of the paper is clear and well supported by figures and tables.

4. While the improvements are modest in certain cases, the results consistently surpass strong baselines, indicating that the approach has potential significance for future research in graph–LLM integration.

**Weaknesses:**

1. The framework primarily combines JEPA with a Multi-Head Attention–based node sampler, which appears somewhat ad hoc and lacks sufficient theoretical justification for why this combination effectively addresses graph misalignment. In addition, the designed modules are not adequately demonstrated, leaving readers uncertain about the necessity of each module like Concept Encoder-Decoder in supporting graph representation.

2. In terms of experiments, the evaluation is restricted to relatively small frozen LLMs (1B–13B), leaving it unclear whether the approach would remain effective or scalable for larger models, or if it is only effective for specialized small-scale models.

3. The evaluation metrics are rather limited, focusing solely on Accuracy and Hit@1. Incorporating more fine-grained measures (e.g., F1, Recall@k, Hit@k, or efficiency-related metrics) would enable a more comprehensive assessment of the proposed approach.

4. Furthermore, the readability of the paper could be improved, as the excessive use of subscript abbreviations makes the technical sections difficult to follow. In addition, there is a typo in the explanation of Equation (1), where the symbol for the text-modal knowledge T is missing. Also, the explanation of Table omits discussion of the CWQ dataset.

**Questions:**

Could the authors clarify the role of each module in the AGE framework, such as the Concept Encoder-Decoder, and explain how they contribute to the overall graph representations through additional descriptions or supporting experiments?

How is the training process in terms of complexity and duration? After training, is the inference time comparable to that of non-LLM retrieval methods?

Could the authors comment on whether the method would remain effective when applied to larger-scale LLMs, or if it is more suitable for small-scale specialized models?

Could the authors provide additional metrics to better demonstrate the effectiveness of the experiments, and include a deeper analysis of these metrics to reveal the intrinsic mechanisms of the proposed method?

---

> ### Author Response · Authors · 2025-11-15
>
> Thank you for carefully reading our manuscript and providing detailed constructive feedback.
>
> __Q1:__ Could the authors clarify the role of each module in the AGE framework, such as the Concept Encoder-Decoder, and explain how they contribute to the overall graph representations through additional descriptions or supporting experiments?
>
> __Response:__ To explain the motivation for each each module in the AGE framework, we analyze the learning objective for expected $\overline{S^{*}}$ representation for LLM $\pi_{\theta}$ on LoRA finetuning as (__see Supplementary Material A for mathematical notation__):
>
> \begin{aligned}
> \mathcal{L} = \\underbrace{ -\\mathbb{E}\_{ (S^*) } \left[ \log R{\omega} (\overline{S^\*} | S^\* ) \right]  }\_{\\text{Loss of Graph Representation Module}}
> \\; \\;
> \\underbrace{-\\mathbb{E}\_{(q,T^{\*}, \overline{S^\*})}
> \left[\log \pi_{\theta}(i \mid q, T^{\*},\overline{S^{\*}})\,  \pi_{\theta}(r \mid q, T^{\*},\overline{S^\*}, i)\right]
> }\_{\\text{Loss of LLM}}
> \end{aligned}
>
> According to Bayes' Theorem, given an input $X$, a target $Y$, and latent rationales $Z$, we can sample these latent rationales $Z$ from the posterior distribution $P(Z|X, Y)$. This posterior represents the probability of latent $Z$ given both the input $X$ and the target $Y$. To compute the marginal likelihood of obtaining answer $Y$ given input $X$, we marginalize over all possible rationales $Z$:
>
> \begin{align}
> P(Y|X)
> &= \sum_{Z \sim P(Z|X,Y)} P(Z, Y|X)  \\; \\; (a) \\\\
> &= \sum_{Z \sim P(Z|X,Y)} P(Z|X) \cdot P(Y|X, Z) \\; \\; (b)
> \end{align}
>
> The equations above show how to compute the marginal likelihood $P(Y|X)$. Equation (a) makes explicit that $Z$ is sampled from the posterior distribution $P(Z|X,Y)$. Equation (b) applies the chain rule of probability to decompose $P(Z, Y|X)$ into two components: $P(Z|X)$ and $P(Y|X,Z)$.
> Following this analysis, we apply it to the learning objective for target representation $\overline{S^{*}}$ given input $S^{\*}$, latent $\mathcal{Z}$ from a posterior $R{\omega}(\mathcal{Z}|S^{\*}, \overline{S^{\*}})$ that bridges $S^{\*}$ and $\overline{S^{\*}}$. The marginal likelihood of $\overline{S^{\*}}$ given $S^{\*}$ is:
>
>
>
> \begin{aligned}
> R\omega(\overline{S^\*}|S^\*)
> &=\sum_{Z\sim R\omega(Z|S^\*,\overline{S^\*})}R\omega(Z,\overline{S^\*}|S^\*) \\\\
> &=\sum_{Z\sim R\omega(Z|S^\*,\overline{S^\*})}R\omega(Z|S^\*)\cdot R\omega(\overline{S^\*}|S^\*,Z)
> \end{aligned}
>
> Above analysis shows that learning objective Graph Representation implicitly learns to identify the latent $Z$ and map it to the expected $\overline{S^{*}}$ for LLM. The extension of the loss function is:
> \begin{aligned}
> -\mathbb{E}\\!\left[\log_{R{\omega}}(\overline{S^{\*}} \mid S^{\*})  \right]
> \&=
> \\underbrace{
> -\mathbb{E}%_{(S^{\*},\overline{S^{\*}})}
> \\! \\left[ \log R{\omega}(\mathcal{Z} \mid S^{\*},\overline{S^{\*}}) \\right]
> }\_{\\text{Loss of Latent Identification}}
> \\; \\;
> \\underbrace{
> -\mathbb{E}\\!\left[
> \log R{\omega}(\mathcal{Z} \mid S^{\*}) \cdot R{\omega}(\overline{S^{\*}} \mid S^{\*},\mathcal{Z}))
> \right]
> }\_{\\text{Loss of Representation}}
> \end{aligned}
>
> Instead of using a single model for both latent $\mathcal{Z}$ identification and $\overline{S^{*}}$ representation learning. We explicitly separate the learning into $Sampler_{\theta}$ for latent identification and $Encoder_{\theta}$-$Decoder_{\theta}$ for representation as:
>
> \begin{aligned}
> -\mathbb{E} \\! \left[\log_{R{\omega}}(\overline{S^{\*}} \mid S^{\*})  \right]
> &\approx
> \\underbrace{
> -\mathbb{E}_{(S^{\*}, \overline{S^{\*}})} \\!
> \left[
> V\_{\\text{key}} \\in \\mathcal{Z} \sim \log \\text{Sampler}\_\theta(\mathcal{Z} \mid S^{\*},\overline{S^{\*}})
> \right]
> }\_{\\text{Loss of Node Sampling}} \\\\ \&
> \\underbrace{
> -\mathbb{E}\_{ ( ( \text{V}\_{key} \\; \, \\;  \Delta\_{V\_{masked}}) \\in S^{\*} )}
> \\! \left[
> \log \\text{Encoder}\_{\theta} (\mathcal{Z} \mid \text{V}\_{key}) \cdot \\text{Decoder}\_{\theta}  (\overline{S^{\*}} \\mid \Delta\_{V\_{masked}},\mathcal{Z})
> \right]
> }\_{\\text{Loss of Encoder-Decoder}}
> \end{aligned}
>
> By separating the learning processes, the target encoder learns the representation directly, while the encoder-decoder learns to reconstruct this representation through __Evidence Lower Bound (ELBO)__ optimization. Specifically, the node sampler learns to extrapolate $V_{key} \in \mathcal{Z}$, rendering static sampling illogical. Therefore, our node sampler with encoder-decoder architecture and explicit loss distribution yields efficient learning signals, faster convergence, and improved graph representations.
>
> To support our analysis, we provide empirical comparison experiments on sampling strategies in Section B.3.7 of the Supplementary Material, and an evaluation of the stability of the target encoder (teacher module) for the driven encoder-decoder in Section C.3.13 of the Supplementary Material.

---

> ### Author Response · Authors · 2025-11-15
>
> __Q2:__ How is the training process in terms of complexity and duration? After training, is the inference time comparable to that of non-LLM retrieval methods?
>
> __Response:__ As shown in Section B.3.4 of the Supplementary Material, we compare inference time between the a non-LLM retrieval prompt-tuned LLM method and baseline. The results indicate that AGE achieves superior performance while maintaining the baseline inference time and remains comparable to the non-retrieval prompt-tuned LLM approach.
>
> __Q3:__ Could the authors comment on whether the method would remain effective when applied to larger-scale LLMs, or if it is more suitable for small-scale specialized models?
>
> __Response:__ Our approach is fundamentally aimed at improving the performance of small-scale models, making them applicable to various KGQA scenarios. We agree that extending our evaluation to assess effectiveness when applied to larger-scale LLMs would be a valuable direction for future work.
>
>
> __Q4:__ Could the authors provide additional metrics to better demonstrate the effectiveness of the experiments, and include a deeper analysis of these metrics to reveal the intrinsic mechanisms of the proposed method?
>
> __Response:__ Thank you for suggesting these additional metrics to strengthen our work. We have updated the evaluation to include F1 and Acc. Following previous work, we also assess the structural quality of generated S-expressions using two additional metrics: Exact Match ratio (EM) and Match-after-Beam-Search ratio (BM), both compared against baseline methods.
>
> \\begin{array}{|l|c|c|c|c|c|}
> \hline
> \\textbf{Method} & \\textbf{F1} & \\textbf{Hits@1} & \\textbf{Acc} & \\textbf{EM} & \\textbf{BM} \\\\
> \\hline
> \\text{AMAR LLaMA2-7B} & 81.2 & 84.3 & 75.2 & 63.9 & 76.4 \\\\
> \\text{AGE AMAR LLaMA2-7B} & 83.7 & 86.5 & 82.2 & 78.6 & 82.0 \\\\
> \\hline
> \\end{array}
>
> From the table above, we observe that the proposed method significantly improves Acc, EM, and BM compared to AMAR. This indicates that AGE enhances both exact matching and beam-search matching, leading to stronger reasoning consistency and better semantic alignment. Our approach prioritizes reasoning and structural, which explains the substantial gains in EM and BM while metrics such as F1 and Hits@1 are more sensitive to answer coverage and ranking ranking—objectives that are secondary in our current design.

---

> > ### Author Response · Authors · 2025-11-28
> >
> > Dear Reviewer r4Hw,
> >
> > We hope this message finds you well.
> >
> > As the discussion period ends is approaching, we would greatly appreciate it if you could kindly acknowledge and respond to our rebuttal for Submission 6889. Thank you again for your comments and suggestions to improve our paper, and we look forward to your reply.
> >
> > Best regards, Authors of Submission 6889

---

### Official Review · Reviewer_mUCY · 2025-10-31

**Soundness:** 2
**Presentation:** 2
**Contribution:** 2
**Rating:** 4
**Confidence:** 3

**Summary:**

This paper introduces AGE (Adaptive-masking for Graph Embedding), a novel node-masking self-supervised learning strategy that adapts masking patterns for improved graph representation, specifically designed for integration with GraphRAG (Graph Retrieval-Augmented Generation) systems that interface with large language models (LLMs). AGE replaces random masking with a reinforcement learning-based node sampler targeting key nodes in the graph, aiming to improve the alignment between graph structure and textual representations, particularly for frozen LLMs with non-parametric retrievers. The methodology integrates a JEPA-inspired encoder-predictor-target architecture and demonstrates consistent accuracy gains on several GraphQA benchmarks.

**Strengths:**

- Clear Motivation and Problem Statement: The paper clearly states the challenge of misalignment between graph and text embeddings for LLMs, especially when using frozen LLMs and non-parametric retrievers. The rationale for focusing on masking non-key nodes is convincing.


- Novelty in Masking Strategy: Introducing a reinforcement learning-based node sampler to select key nodes for masking represents a creative adaptation over random masking, and is conceptually well-motivated by the concise, non-redundant nature of graphs.


- Comprehensive Experimental Results: The experimental results and ablation studies offer multi-perspective evaluation across multiple datasets and model settings. The experimental design is careful, and performance gains are significant; in particular, AGE shows 26.7 percentage point improvement on ExplaGraphs compared to the G-Retriever baseline with Llama3.2-1B.

**Weaknesses:**

+ Inadequate Ablation on the Node Sampler's Effectiveness on Other Tasks: The paper only tests AGE on GraphRAG (GraphQA) tasks. As stated in the limitations, it is unclear whether the RL-based node sampler for adaptive masking would retain its efficacy for other graph embedding applications (e.g., node classification, link prediction, or other modalities). Targeted experiments on at least one such task would clarify broader utility.
+ Lack of Theoretical Analysis for Node Sampler: While the reinforcement learning/REINFORCE formulation is presented, there is little theoretical or empirical explanation as to why the learned node saliency effectively distinguishes 'key' from 'auxiliary' nodes in a way that generalizes or improves graph representation universally. Are there common node properties (e.g., centrality, clustering, bottlenecks) that the sampler converges to in practice? The color-coded qualitative t-SNE in Figure 5 is informative but anecdotal.
+ Lack of comparison with strong heuristic baselines for key-node selection: The paper does not compare the learnable node sampler with several simple yet reasonable heuristic strategies (e.g., degree-based, PageRank, betweenness centrality, TF-IDF for textual importance, or top-k nodes ranked by retrieval scores). If such heuristic methods already provide similar improvements, the additional complexity of a learnable sampler may not be sufficiently justified. Currently, the paper only contrasts random masking with the learnable sampler, without including these stronger heuristic baselines, which weakens the empirical claim of the proposed module’s necessity.
+ Result Presentation and Statistical Robustness: In Table 1 and 2, no variances or confidence intervals are reported, nor is there discussion of statistical significance or stability across multiple experimental runs. This raises uncertainty regarding the robustness of claimed gains, especially since improvements on some datasets (e.g., WebQSP) are much smaller.

**Questions:**

1. Could you please provide more rigorous analysis (theoretical or empirical) of how the RL-based node sampler distinguishes key nodes from non-key nodes? For example, are there emergent properties, metrics, or correlations with known graph-theoretic quantities?

2. Have you run any of the ablation tests or main experiments on graph-centric tasks outside the GraphQA/GraphRAG scope (e.g., node classification, link prediction) using public benchmarks? If so, please provide results or commentary.

3. Could you systematically report variance or statistical significance (multiple seeds/runs) for key results, especially those with less dramatic improvements?

4. Is the node sampler robust to varying graph sizes/densities, or is model performance highly sensitive to the fixed sampling rate?

---

> ### Author Response · Authors · 2025-11-15
>
> Thank you for carefully reading our manuscript and providing detailed and valuable feedback. Below we respond to your questions and concerns.
>
> __Q1:__ Could you please provide more rigorous analysis (theoretical or empirical) of how the RL-based node sampler distinguishes key nodes from non-key nodes? For example, are there emergent properties, metrics, or correlations with known graph-theoretic quantities?
>
> __Response:__ To explain how RL-based node sampler distinguishes key nodes from $S^{\*}$. We analyze the learning objective for expected $\overline{S^{*}}$ representation for LLM $\pi_{\theta}$ on LoRA finetuning as (__see Supplementary Material A for mathematical notation__):
>
> \begin{aligned}
> \mathcal{L} = \\underbrace{ -\\mathbb{E}\_{ (S^*) } \left[ \log R{\omega} (\overline{S^\*} | S^\* ) \right]  }\_{\\text{Loss of Graph Representation Module}}
> \\; \\;
> \\underbrace{-\\mathbb{E}\_{(q,T^{\*}, \overline{S^\*})}
> \left[\log \pi_{\theta}(i \mid q, T^{\*},\overline{S^{\*}})\,  \pi_{\theta}(r \mid q, T^{\*},\overline{S^\*}, i)\right]
> }\_{\\text{Loss of LLM}}
> \end{aligned}
>
> According to Bayes' Theorem, given an input $X$, a target $Y$, and latent rationales $Z$, we can sample these latent rationales $Z$ from the posterior distribution $P(Z|X, Y)$. This posterior represents the probability of latent $Z$ given both the input $X$ and the target $Y$. To compute the marginal likelihood of obtaining answer $Y$ given input $X$, we marginalize over all possible rationales $Z$:
>
> \begin{align}
> P(Y|X)
> &= \sum_{Z \sim P(Z|X,Y)} P(Z, Y|X)  \\; \\; (a) \\\\
> &= \sum_{Z \sim P(Z|X,Y)} P(Z|X) \cdot P(Y|X, Z) \\; \\; (b)
> \end{align}
>
> The equations above show how to compute the marginal likelihood $P(Y|X)$. Equation (a) makes explicit that $Z$ is sampled from the posterior distribution $P(Z|X,Y)$. Equation (b) applies the chain rule of probability to decompose $P(Z, Y|X)$ into two components: $P(Z|X)$ and $P(Y|X,Z)$.
> Following this analysis, we apply it to the learning objective for target representation $\overline{S^{*}}$ given input $S^{\*}$, latent $\mathcal{Z}$ from a posterior $R{\omega}(\mathcal{Z}|S^{\*}, \overline{S^{\*}})$ that bridges $S^{\*}$ and $\overline{S^{\*}}$. The marginal likelihood of $\overline{S^{\*}}$ given $S^{\*}$ is:
>
>
>
> \begin{aligned}
> R\omega(\overline{S^\*}|S^\*)
> &=\sum_{Z\sim R\omega(Z|S^\*,\overline{S^\*})}R\omega(Z,\overline{S^\*}|S^\*) \\\\
> &=\sum_{Z\sim R\omega(Z|S^\*,\overline{S^\*})}R\omega(Z|S^\*)\cdot R\omega(\overline{S^\*}|S^\*,Z)
> \end{aligned}
>
> Above analysis shows that learning objective Graph Representation implicitly learns to identify the latent $Z$ and map it to the expected $\overline{S^{*}}$ for LLM. The extension of the loss function is:
> \begin{aligned}
> -\mathbb{E}\\!\left[\log_{R{\omega}}(\overline{S^{\*}} \mid S^{\*})  \right]
> \&=
> \\underbrace{
> -\mathbb{E}%_{(S^{\*},\overline{S^{\*}})}
> \\! \\left[ \log R{\omega}(\mathcal{Z} \mid S^{\*},\overline{S^{\*}}) \\right]
> }\_{\\text{Loss of Latent Identification}}
> \\; \\;
> \\underbrace{
> -\mathbb{E}\\!\left[
> \log R{\omega}(\mathcal{Z} \mid S^{\*}) \cdot R{\omega}(\overline{S^{\*}} \mid S^{\*},\mathcal{Z}))
> \right]
> }\_{\\text{Loss of Representation}}
> \end{aligned}
>
> Instead of using a single model for both latent $\mathcal{Z}$ identification and $\overline{S^{*}}$ representation learning. We explicitly separate the learning into $Sampler_{\theta}$ for latent identification and $Encoder_{\theta}$-$Decoder_{\theta}$ for representation as:
>
> \begin{aligned}
> -\mathbb{E}\\!\left[\log_{R{\omega}}(\overline{S^{\*}} \mid S^{\*})  \right]
> &\approx
> \\underbrace{
> -\mathbb{E}\_{(S^{\*}, \overline{S^{\*}})} \\!
> \left[
> V\_{\\text{key}} \\in \\mathcal{Z} \sim \log \\text{Sampler}\_\theta(\mathcal{Z} \mid S^{\*},\overline{S^{\*}})
> \right]
> }\_{\\text{Loss of Node Sampling}} \\\\ \&
> \\underbrace{
> -\mathbb{E}\_{ ( ( \text{V}\_{key} \\; \, \\;  \Delta\_{V\_{masked}}) \\in S^{\*} )}
> \\! \left[
> \log \\text{Encoder}\_{\theta} (\mathcal{Z} \mid \text{V}\_{key}) \cdot \\text{Decoder}\_{\theta}  (\overline{S^{\*}} \\mid \Delta\_{V\_{masked}},\mathcal{Z})
> \right]
> }\_{\\text{Loss of Encoder-Decoder}}
> \end{aligned}
> By separating the learning processes, the target encoder learns the representation directly, while the encoder-decoder learns to reconstruct this representation through __Evidence Lower Bound (ELBO)__ optimization. Specifically, the node sampler learns to extrapolate $V_{key} \in \mathcal{Z}$, rendering static sampling illogical. Therefore, our node sampler with encoder-decoder architecture and explicit loss distribution yields efficient learning signals, faster convergence, and improved graph representations.
>
> To support our analysis, we provide empirical comparison experiments on sampling strategies in Section B.3.7 of the Supplementary Material, and an evaluation of the stability of the target encoder (teacher module) for the driven encoder-decoder in Section C.3.13 of the Supplementary Material.

---

> ### Author Response · Authors · 2025-11-15
>
> __Q2:__ Have you run any of the ablation tests or main experiments on graph-centric tasks outside the GraphQA/GraphRAG scope (e.g., node classification, link prediction) using public benchmarks? If so, please provide results or commentary.
>
> __Response:__ Thank you for raising this important point. While additional dataset evaluations would further strengthen our findings, our approach is inherently dataset-agnostic and applicable to diverse KGQA scenarios. Moreover, tasks such as node classification and link prediction are fully compatible with our method. We acknowledge that extending our evaluation to these datasets would be a valuable direction for future work.
>
> __Q3:__ Could you systematically report variance or statistical significance (multiple seeds/runs) for key results, especially those with less dramatic improvements?
>
> __Response:__ We appreciate the reviewer’s insightful observation. Below, we present experimental results validated across seeds 0~4, showing that our approach consistently outperforms the baseline method.
>
>
> \\begin{array}{|l|c|c|c|}
> \hline
> \\textbf{Method} & \\textbf{ExplaGraphs} & \\textbf{SceneGraphs} & \\textbf{WebQSP} \\\\
> \\hline
> \\text{G-Retriever Llama3.2-1B} & 0.5595 & 0.7540 & 60.1 \\\\
> \\text{G-Retriever Llama3.2-3B} & 0.7761 & 0.8229 & 71.3 \\\\
> \\text{G-Retriever Llama2-7B}   & 0.8516 & 0.8131 & 68.1 \\\\
> \\hline
> \\text{AGE G-Retriever Llama3.2-1B} & 0.8214 \\pm 0.020 & 0.8112 \\pm 0.070 & 61.5 \\pm 1.2 \\\\
> \\text{AGE G-Retriever Llama3.2-3B} & 0.9186 \\pm 0.019 & 0.8874 \\pm 0.089 & 72.7 \\pm 1.5 \\\\
> \\text{AGE G-Retriever Llama3.2-3B} & 0.9286 \\pm 0.033 & 0.9244 \\pm 0.033 & 77.5 \\pm 1.3 \\\\
> \\hline
> \\text{G-Retriever Llama3.2-1B-LoRA} & 0.7328 & 0.8689 & 65.3 \\\\
> \\text{G-Retriever Llama3.2-3B-LoRA} & 0.8339 & 0.9074 & 71.4 \\\\
> \\text{G-Retriever Llama2-7B-LoRA}   & 0.8705 & 0.8683 & 70.2 \\\\
> \\hline
> \\text{AGE G-Retriever Llama3.2-1B-LoRA} & 0.8433 \pm 0.0089 & 0.9052 \pm 0.002 & 68.6 \pm 1.4 \\\\
> \\text{AGE G-Retriever Llama3.2-3B-LoRA} & 0.9243 \pm 0.0202 & 0.9439 \pm 0.007 & 75.9 \pm 1.1 \\\\
> \\text{AGE G-Retriever Llama3.2-3B-LoRA} & 0.9549 \pm 0.0113 & 0.9256 \pm 0.003 & 79.5 \pm 2.0 \\\\
> \\hline
> \\end{array}
>
> __Q4:__ Is the node sampler robust to varying graph sizes/densities, or is model performance highly sensitive to the fixed sampling rate?
>
> __Response:__ Thank you for the insightful question. While AGE leverage tuned sampling rates in certain settings, our extensive experiments indicate that its robust performance primarily stems from significant improvements in retrieval.
> As shown in Section B.3.9 of the Supplementary Material, retrieval recall values are reported across varying retrieval sizes. The results demonstrate that AGE consistently outperforms baseline method. Notably, AGE maintains strong performance even with small retrieval sizes, whereas baselines require larger sizes to achieve effective representation learning. This property is particularly valuable for enhancing the reasoning capabilities of LLMs in contexts with limited retrieval resources.

---

> > ### Author Response · Authors · 2025-11-28
> >
> > Dear Reviewer mUCY,
> >
> > We hope this message finds you well.
> >
> > As the discussion period ends is approaching, we would greatly appreciate it if you could kindly acknowledge and respond to our rebuttal for Submission 6889. Thank you again for your comments and suggestions to improve our paper, and we look forward to your reply.
> >
> > Best regards, Authors of Submission 6889

---

### Official Review · Reviewer_wmUU · 2025-11-01

**Soundness:** 4
**Presentation:** 2
**Contribution:** 3
**Rating:** 4
**Confidence:** 3

**Summary:**

This paper proposes AGE, an adaptive-masking embedding strategy for graph designed to enhance Graph Retrieval-Augmented Generation (GraphRAG) for Large Language Model (LLM) by improving the alignment and quality of graph-structured embeddings for frozen LLM through a self-supervised learning approach that uses a reinforcement learning-guided node sampler to strategically mask auxiliary nodes instead of critical key nodes.

**Strengths:**

1. The central concept of adaptively masking nodes in a graph, rather than using random masking, is well-motivated for graph structure data. By masking auxiliary nodes during SSL process for graph embedding module, LLMs can identify redundant information within retrieved graph.
2. The overall architecture is well designed. It integrates the Joint-Embedding Predictive Architecture into the GraphRAG context, moving beyond simple generative or contrastive approaches, to alleviating the reconstruction noise and collapse risk of generative SSL. And the use of a Reinforcement Learning based node sampler to solve the non-differentiable selection problem is a clever and appropriate technical choice.
3. AGE is a practical method for improving GraphRAG performance. It update graph embedding module on non-parametric retrievers with frozen LLMs, which avoids end‑to‑end LLM fine‑tuning, and keeps compute within reasonable bounds for real deployments.

**Weaknesses:**

1. How well the graph embedding module can be under AGE framework depends largely on whether the node sampler can find out key nodes within a graph. Currently, AGE use a trainable MHA network to predict key nodes. And it's better to analyze the performance between different node sample strategy and see whether it would affect the model performance, which can provide more compelling evidence for the core insight about key nodes of this paper.
2. The training pipeline is too complex under three different loss function and lots of modules, which may leads to unstable training process for AGE. And the authors do not provide more detail about the experiments setup and implementation details about the training process. How to make so many modules converge as quickly as possible during the training process?
3. The work claims that AGE can be used for arbitrary LLMs, but experiments are all conducted under Llama series model. It would be better to incorporate other LLMs such as Qwen in the experiments to further confirm this claim.

**Questions:**

1. Although there is no overlap between the three losses, is it really feasible to directly sum the three loss functions? This is somewhat counterintuitive.

---

> ### Author Response · Authors · 2025-11-15
>
> Thank you for your insightful and positive feedback. Below we respond to your questions and concerns.
>
> __Questions:__  Although there is no overlap between the three losses, is it really feasible to directly sum the three loss functions? This is somewhat counterintuitive.
>
> __Response:__ To explain the motivation for architecture design and applying distributed loss on each module, we analyze the learning objective for expected $\overline{S^{*}}$ representation for LLM $\pi_{\theta}$ on LoRA finetuning as (__see Supplementary Material A for mathematical notation__):
>
> \begin{aligned}
> \mathcal{L} = \\underbrace{ -\\mathbb{E}\_{ (S^*) } \left[ \log R{\omega} (\overline{S^\*} | S^\* ) \right]  }\_{\\text{Loss of Graph Representation Module}}
> \\; \\;
> \\underbrace{-\\mathbb{E}\_{(q,T^{\*}, \overline{S^\*})}
> \left[\log \pi_{\theta}(i \mid q, T^{\*},\overline{S^{\*}})\,  \pi_{\theta}(r \mid q, T^{\*},\overline{S^\*}, i)\right]
> }\_{\\text{Loss of LLM}}
> \end{aligned}
>
> According to Bayes' Theorem, given an input $X$, a target $Y$, and latent rationales $Z$, we can sample these latent rationales $Z$ from the posterior distribution $P(Z|X, Y)$. This posterior represents the probability of latent $Z$ given both the input $X$ and the target $Y$. To compute the marginal likelihood of obtaining answer $Y$ given input $X$, we marginalize over all possible rationales $Z$:
>
> \begin{align}
> P(Y|X)
> &= \sum_{Z \sim P(Z|X,Y)} P(Z, Y|X)  \\; \\; (a) \\\\
> &= \sum_{Z \sim P(Z|X,Y)} P(Z|X) \cdot P(Y|X, Z) \\; \\; (b)
> \end{align}
>
> The equations above show how to compute the marginal likelihood $P(Y|X)$. Equation (a) makes explicit that $Z$ is sampled from the posterior distribution $P(Z|X,Y)$. Equation (b) applies the chain rule of probability to decompose $P(Z, Y|X)$ into two components: $P(Z|X)$ and $P(Y|X,Z)$.
> Following this analysis, we apply it to the learning objective for target representation $\overline{S^{*}}$ given input $S^{\*}$, latent $\mathcal{Z}$ from a posterior $R{\omega}(\mathcal{Z}|S^{\*}, \overline{S^{\*}})$ that bridges $S^{\*}$ and $\overline{S^{\*}}$. The marginal likelihood of $\overline{S^{\*}}$ given $S^{\*}$ is:
>
>
>
> \begin{aligned}
> R\omega(\overline{S^\*}|S^\*)
> &=\sum_{Z\sim R\omega(Z|S^\*,\overline{S^\*})}R\omega(Z,\overline{S^\*}|S^\*) \\\\
> &=\sum_{Z\sim R\omega(Z|S^\*,\overline{S^\*})}R\omega(Z|S^\*)\cdot R\omega(\overline{S^\*}|S^\*,Z)
> \end{aligned}
>
> Above analysis shows that learning objective Graph Representation implicitly learns to identify the latent $Z$ and map it to the expected $\overline{S^{*}}$ for LLM. The extension of the loss function is:
> \begin{aligned}
> -\mathbb{E}\\!\left[\log_{R{\omega}}(\overline{S^{\*}} \mid S^{\*})  \right]
> \&=
> \\underbrace{
> -\mathbb{E}%_{(S^{\*},\overline{S^{\*}})}
> \\! \\left[ \log R{\omega}(\mathcal{Z} \mid S^{\*},\overline{S^{\*}}) \\right]
> }\_{\\text{Loss of Latent Identification}}
> \\; \\;
> \\underbrace{
> -\mathbb{E}\\!\left[
> \log R{\omega}(\mathcal{Z} \mid S^{\*}) \cdot R{\omega}(\overline{S^{\*}} \mid S^{\*},\mathcal{Z}))
> \right]
> }\_{\\text{Loss of Representation}}
> \end{aligned}
>
> Instead of using a single model for both latent $\mathcal{Z}$ identification and $\overline{S^{*}}$ representation learning. We explicitly separate the learning into $Sampler_{\theta}$ for latent identification and $Encoder_{\theta}$-$Decoder_{\theta}$ for representation as:
>
> \begin{aligned}
> -\mathbb{E}\\!\left[\log_{R{\omega}}(\overline{S^{\*}} \mid S^{\*})  \right]
> &\approx
> \\underbrace{
> -\mathbb{E}\_{(S^{\*}, \overline{S^{\*}})} \\!
> \left[
> V\_{\\text{key}} \\in \\mathcal{Z} \sim \log \\text{Sampler}\_\theta(\mathcal{Z} \mid S^{\*},\overline{S^{\*}})
> \right]
> }\_{\\text{Loss of Node Sampling}} \\\\ \&
> \\underbrace{
> -\mathbb{E}\_{ ( ( \text{V}\_{key} \\; \, \\;  \Delta\_{V\_{masked}}) \\in S^{\*} )}
> \\! \left[
> \log \\text{Encoder}\_{\theta} (\mathcal{Z} \mid \text{V}\_{key}) \cdot \\text{Decoder}\_{\theta}  (\overline{S^{\*}} \\mid \Delta\_{V\_{masked}},\mathcal{Z})
> \right]
> }\_{\\text{Loss of Encoder-Decoder}}
> \end{aligned}
>
> By separating the learning processes, the target encoder learns the representation directly, while the encoder-decoder learns to reconstruct this representation through __Evidence Lower Bound (ELBO)__ optimization. Specifically, the node sampler learns to extrapolate $V_{key} \in \mathcal{Z}$, rendering static sampling illogical. Therefore, our node sampler with encoder-decoder architecture and explicit loss distribution yields efficient learning signals, faster convergence, and improved graph representations.
>
> To support our analysis, we provide empirical comparison experiments on sampling strategies in Section B.3.7 of the Supplementary Material, and an evaluation of the stability of the target encoder (teacher module) for the driven encoder-decoder in Section C.3.13 of the Supplementary Material.

---

> > ### Comment · Reviewer_wmUU · 2025-11-27
> >
> > Thank you for your response and detailed introduction to the motivation for the architecture design and the application of distributed loss to each module, which partially addresses my concern. Although my question was not directly answered, I appreciate your explanation.
> >
> > However, **most of my concerns are listed in the weaknesses section of my review** . Could you please provide further explanation or clarification regarding these points?

---

> ### Author Response · Authors · 2025-11-27
>
> Thank you for your response and for continuing this discussion. Below are my comments and concerns:
>
> __Q1:__ Currently, AGE use a trainable MHA network to predict key nodes. And it's better to analyze the performance between different node sample strategy and see whether it would affect the model performance, which can provide more compelling evidence for the core insight about key nodes of this paper.
>
> __Response:__ We agree that the effectiveness of the graph embedding module under the AGE framework is closely tied to the ability of the node sampler to identify key nodes. To address this, we have already included comprehensive comparisons of different node sampling strategies in Supplementary Material __(see Supplementary Material Section B.3.7, Figure C)__.
>
> Following the above discussion, the extension of the loss function is:
> \begin{aligned}
> -\mathbb{E}\\!\left[\log_{R{\omega}}(\overline{S^{\*}} \mid S^{\*})  \right]
> \&=
> \\underbrace{
> -\mathbb{E}%_{(S^{\*},\overline{S^{\*}})}
> \\! \\left[ \log R{\omega}(\mathcal{Z} \mid S^{\*},\overline{S^{\*}}) \\right]
> }\_{\\text{Loss of Latent Identification}}
> \\; \\;
> \\underbrace{
> -\mathbb{E}\\!\left[
> \log R{\omega}(\mathcal{Z} \mid S^{\*}) \cdot R{\omega}(\overline{S^{\*}} \mid S^{\*},\mathcal{Z}))
> \right]
> }\_{\\text{Loss of Representation}}
> \end{aligned}
>
> This extension illustrates how our design enables the module to learn latent identification. We employ reinforcement learning (RL) to estimate latent variable $\mathcal{Z}$ based on input $S^{\*}$ and expected output $\overline{S^{\*}}$ nodes—a capability that static sampling methods lack. To support this claim, we provide empirical comparisons of sampling strategies in __Section B.3.7 of the Supplementary Material__.
>
> Our findings show that static strategies such as PageRank and Degree Centrality require higher sampling rates to achieve high performance. This suggests that key nodes identified by static methods are less impactful because the encoder-decoder implicitly learns its own identification strategy, necessitating a larger set of nodes for effective graph embedding.
> In contrast, RL-based node samplers achieve strong performance even with lower sampling rates. This indicates that RL-selected key nodes better support the encoder-decoder, leading to higher-quality graph embeddings and enabling the LLM to produce accurate answers with fewer guiding nodes.
>
> __Q2:__ The training pipeline is too complex under three different loss function and lots of modules, which may leads to unstable training process for AGE. And the authors do not provide more detail about the experiments setup and implementation details about the training process. How to make so many modules converge as quickly as possible during the training process?
>
> __Response:__ We acknowledge that incorporating multiple loss functions and modules can introduce complexity and potential instability during training. To address this, we conducted a study on two widely used techniques that improve the stability of the concept encoder-decoder, as detailed in __Supplementary Material Section B.3.13 and Table H__.
>
> Specifically, we explored Exponential Moving Average (EMA) and normalization. EMA maintains a smoothed version of encoder weights, reducing fluctuations, while normalization ensures consistent activation distributions and mitigates internal covariate shift. Our results show that normalization significantly improves encoder stability and performance, and EMA provides additional gains. Based on these findings, we applied normalization in all experiments but omitted EMA for simplicity.
>
> In addition, we explicitly separate the roles of each module and distribute loss across modules to provide efficient learning signals, which accelerates convergence overall training.
>
> For comprehensive details on the experimental setup and training implementation, please refer to __Supplementary Material Sections B.1 and B.2.__
>
>
> __Q3:__ The work claims that AGE can be used for arbitrary LLMs, but experiments are all conducted under Llama series model. It would be better to incorporate other LLMs such as Qwen in the experiments to further confirm this claim.
>
> __Response:__ Thank you for your valuable suggestion. We agree that evaluating AGE on a broader range of LLMs, such as Qwen, would further strengthen the claim of its general applicability. In our current experiments, we followed the common practice in prior work by using the Llama series to ensure comparability with existing results, as it is widely adopted in the community. We acknowledge the importance of demonstrating AGE’s effectiveness across diverse architectures. As part of future work, we extend our evaluation to include additional models like Qwen to provide a more comprehensive validation of AGE’s generality.

---

### Meta-Review · Area_Chair_Yp7M · 2025-12-10

**Summary:**

This paper proposes the AGE framework, aiming to improve the alignment between the graph structure and text representation of frozen LLM in the GraphRAG environment. Its core innovation lies in combining the JEPA with a reinforcement learning-based node sampler, selectively masking auxiliary nodes to help the model learn more robust representations of key nodes.

Reviewers generally agreed that the research motivation was clear and the problem of graph-text misalignment was well-expressed. They praised the reinforcement learning-based adaptive masking strategy for its greater innovation compared to random masking. However, common concerns included the complexity of the architecture, the magnitude of empirical gains, the adequacy of experimental validation (metrics, baselines, and tasks), and the theoretical basis of the proposed modules, which prevented the paper from meeting the ICLR acceptance criteria.

**Reviewer Concerns:**

### Addressed Concerns:

- Reviewers **wmUU** questioned whether the learnable sampler was better than other sampling strategies. The authors pointed to specific ablation studies in the Appendix (B.3.7) comparing their method against static strategies like PageRank and Degree Centrality, showing the RL sampler's superiority at lower sampling rates.
- Reviewer **mUCY** raised concerns about the lack of error bars. The authors addressed this by providing results with standard deviations across 5 seeds (0-4), demonstrating consistent improvements.
- Reviewer **r4Hw** noted the limitation of only using Accuracy and Hit@1. The authors responded by adding F1, EM, and BM metrics, which showed favorable results for their method.
- Reviewers **wmUU** and **r4Hw** questioned the rationale behind the complex architecture (summing losses, specific modules). The authors provided a detailed mathematical analysis deriving their learning objectives from Bayes' Theorem to justify the separation of latent identification (Sampler) and representation learning (Encoder-Decoder).
- Reviewer **J96N** asked for evidence that the model actually "reasons" better. The authors highlighted visualizations in the Appendix showing that their method clusters key nodes (true answers) separately from incorrect answers.

### Outstanding Concerns:

The authors did not address the concerns raised by the reviewers regarding weaknesses, including the recent baseline methods, non-QA tasks, scalability of larger LLMs, cost-benefit trade-offs, etc.

**Reviewer Scores:**

Given that the authors did not clarify the weaknesses that the reviewers were more concerned about, the reviewers are unlikely to change their opinions. The score may remain unchanged (4444).

---

### Decision · Program_Chairs · 2026-01-26

Reject